# Solar-driven highly selective conversion of glycerol to dihydroxyacetone using surface atom engineered BiVO$_4$ photoanodes

Yuan Lu[1,2,6], Byoung Guan Lee[3,6], Cheng Lin[1], Tae-Kyung Liu[2], Zhipeng Wang[4], Jiaming Miao[1], Sang Ho Oh [5], Ki Chul Kim [3] ✉, Kan Zhang [1] ✉ & Jong Hyeok Park [2] ✉

Dihydroxyacetone is the most desired product in glycerol oxidation reaction because of its highest added value and large market demand among all possible oxidation products. However, selectively oxidative secondary hydroxyl groups of glycerol for highly efficient dihydroxyacetone production still poses a challenge. In this study, we engineer the surface of BiVO$_4$ by introducing bismuth-rich domains and oxygen vacancies (Bi-rich BiVO$_{4-x}$) to systematically modulate the surface adsorption of secondary hydroxyl groups and enhance photo-induced charge separation for photoelectrochemical glycerol oxidation into dihydroxyacetone conversion. As a result, the Bi-rich BiVO$_{4-x}$ increases the glycerol oxidation photocurrent density of BiVO$_4$ from 1.42 to 4.26 mA cm$^{-2}$ at 1.23 V vs. reversible hydrogen electrode under AM 1.5 G illumination, as well as the dihydroxyacetone selectivity from 54.0% to 80.3%, finally achieving a dihydroxyacetone production rate of 361.9 mmol m$^{-2}$ h$^{-1}$ that outperforms all reported values. The surface atom customization opens a way to regulate the solar-driven organic transformation pathway toward a carbon chain-balanced product.

Glycerol oxidation reaction (GOR) has garnered significant attention in recent years, due to its economic advantages stemming from low raw material costs (~ US $0.11 per kg) and the potential to yield a diverse range of products[1–4]. However, previously reported high-efficiency electro-oxidation methods often lead to the breaking of the C-C bond of glycerol, tending to the formation of low-carbon products, such as formic acid (FA)[5–7]. Despite the relatively modest market price of FA (approximately US $0.4 per kg), which places a constraint on the overall potential for value enhancement of GOR[8,9]. The prevalence of primary hydroxyl oxidation in glycerol primarily gives rise to the phenomenon of producing products, featuring asymmetric and delicate functional groups. In contrast, dihydroxyacetone (DHA),

originating from secondary hydroxyl oxidation, yields a three-carbon product with a robust carbon chain. DHA is particularly attractive, driven by substantial market demand and a high market value (approximately US $150 per kg), especially within the cosmetic industry, establishing it as the most coveted product within the GOR[10,11]. As a result, the pursuit of maintaining a balanced carbon chain to selectively produce DHA becomes an appealing endeavor.

Bismuth vanadate (BiVO$_4$) is considered as one of the most potential photoanode materials in photoelectrochemical (PEC) cells because of its suitable energy band position and large light absorption waveband[12,13]. It is essential to highlight that the primary product resulting from glycerol oxidation over the BiVO$_4$ photoanode is DHA.

[1]School of Materials Science and Engineering, Nanjing University of Science and Technology, Nanjing, China. [2]Department of Chemical and Biomolecular Engineering, Yonsei University, 50 Yonsei-ro, Seodaemun-gu, Seoul, Republic of Korea. [3]Computational Materials Design Laboratory, Department of Chemical Engineering, Konkuk University, Seoul, the Republic of Korea. [4]Department of Energy Science, Sungkyunkwan University, Suwon, Republic of Korea. [5]Department of Energy Engineering, Institute for Energy Materials and Devices, Korea Institute of Energy Technology (KENTECH), Naju, Republic of Korea. [6]These authors contributed equally: Yuan Lu, Byoung Guan Lee. ✉e-mail: kich2018@konkuk.ac.kr; zhangkan@njust.edu.cn; lutts@yonsei.ac.kr

However, the modest 50% selectivity of DHA products for BiVO$_4$ proves to be below expectations, leading to inefficient solar conversion in the GOR process[14]. While various strategies, such as optical absorption and charge separation regulation, have been implemented to enhance solar-driven GOR conversion for DHA production[15,16], achieving an improvement in DHA selectivity remains challenging. Recent observations indicate that the bismuth (Bi) atom exhibits a heightened electrostatic adsorption capacity for the secondary hydroxyl of glycerol. This insight suggests that the origin of DHA selectivity is likely associated with the surface Bi atoms of BiVO$_4$[17–19]. Nevertheless, the intrinsic distribution of Bi atoms on the BiVO$_4$ surface does not exert a decisive influence on directing the GOR reaction pathway due to the atomic structure of monoclinic BiVO$_4$, which appears as layered stacking. In this structure, the VO$_4$ unit cell adjacent to the cation restricts the contact of Bi atoms with the electrolyte[20,21]. Hence, the prospect of exposing a greater fraction of Bi atoms on the surface of the BiVO$_4$ photoanode shows significant potential for achieving highly selective DHA production.

Herein, a photoanode film composed of surface Bi-rich BiVO$_4$ particle with mainly exposed the (010) facet is synthesized using a straightforward alkaline immersion method. The Bi-rich BiVO$_4$ exhibits an elevated surface potential along with notably amplified secondary hydroxyl adsorption for glycerol. Furthermore, additional oxygen vacancies (O$_v$) are introduced to enhance the interaction frequency of Bi atoms at the interface, while leading to an improvement in surface charge transport efficiency. Consequently, the photocurrent density of Bi-rich BiVO$_{4-x}$ photoanode in GOR increases from 1.42 to 4.26 mA cm$^{-2}$ at 1.23 V vs. RHE under AM 1.5 G illumination, accompanied by a selectivity increase of DHA product from 54% to 80.3%, finally achieving a DHA conversion of 361.9 mmol m$^{-2}$ h$^{-1}$ that is the highest value so far.

## Results

BiVO$_4$ photoanodes composed of (010) crystal plane exposed micron-sized BiVO$_4$ particles were synthesized by a seed-assisted hydrothermal reaction[22], as determined by field emission scanning electron microscopy (FE-SEM) images in Supplementary Figs. 1–3. The generation of a Bi-enriched surface and the introduction of O$_v$ on the surface of BiVO$_4$ were accomplished through alkali solution etching, followed by sequential electrochemical reduction. Compared to bare BiVO$_4$, both Bi-rich BiVO$_4$ and Bi-rich BiVO$_{4-x}$ have negligible changes in their morphologies (Fig. 1a, b) and crystal structures (Supplementary Fig. 4). Nevertheless, the inductively coupled plasma-mass spectrometry (ICP-MS) results in Supplementary Fig. 5 shows the V atoms being leached from the BiVO$_4$ after alkali solution treatment. To understand where the V in solution after alkali etching comes from, Raman spectra of BiVO$_4$, Bi-rich BiVO$_4$, and Bi-rich BiVO$_{4-x}$ are compared, since the Raman peaks of BiVO$_4$ can be associated with the stretching and vibration of the V-O bond[23]. As shown in Supplementary Fig. 6, the almost unchanged Raman peaks imply that the loss of V after alkali etching does not form V vacancies in BiVO$_4$ crystal, rather which might be mainly originating from surface VO$_4^{3-}$ loss[24]. Correspondingly, high-resolution transmission electron microscopy (HR-TEM) images of the Bi-rich BiVO$_{4-x}$ in Fig. 1c and Supplementary Figs. 7, 8 demonstrate obvious contrast change near the edge and bulk, while element mapping image shows that Bi, V, and O elements are evenly distributed in the grains of Bi-rich BiVO$_{4-x}$, indicating the V loss from the surface shallow region (Supplementary Fig. 9).

To further determine the possible surface component change, electron energy-loss spectroscopy (EELS) was employed. The EELS spectra of BiVO$_4$, Bi-rich BiVO$_4$, and Bi-rich BiVO$_{4-x}$ were extracted linearly from the surface to the bulk region with a 26 nm depth, and the transitions in the fine structure of the vanadium $L_{2,3}$ edge (V-$L_{2,3}$) and the O-K edge at 14 sequential points from the surface to the bulk are shown in Fig. 1d, Supplementary Figs. 10, 11. A high-energy shift of the V-$L_{2,3}$, which is caused by V$^{5+}$ reducing more than two valence states[25,26], can be observed from the surface to the bulk of BiVO$_4$ (Fig. 1e). The energy changes of the vanadium $L_{2,3}$ edge against depth is plotted in Fig. 1f, Supplementary Figs. 10, 11, accordingly. A remarkable

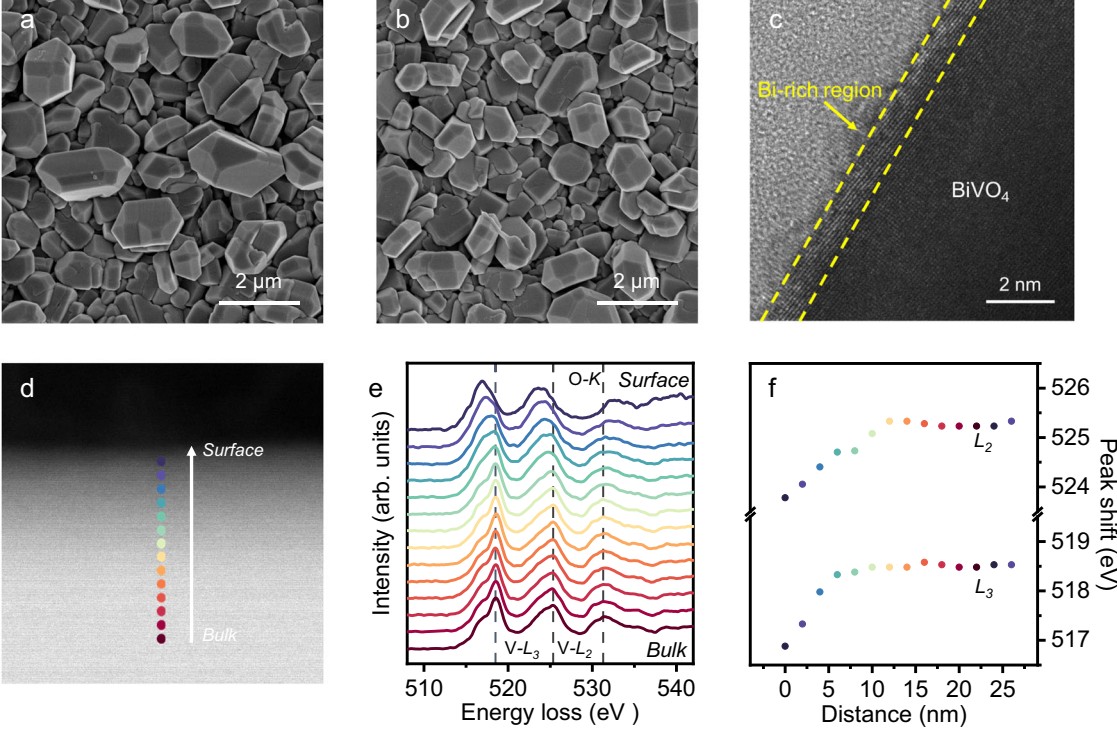

**Fig. 1 | Bi-rich surface construction of BiVO$_4$ photoanode. a, b** Top-view SEM image of Bi-rich BiVO$_4$ and Bi-rich BiVO$_{4-x}$. **c** HR-TEM image of Bi-rich BiVO$_{4-x}$. **d** STEM image of Bi-rich BiVO$_{4-x}$ particle with the probing path shown by the dotted line, scale bar: 2 nm, (**e**) the corresponding EELS spectrum of Bi-rich BiVO$_{4-x}$. **f** the peak shift of V-$L_{2,3}$ edge with depth.

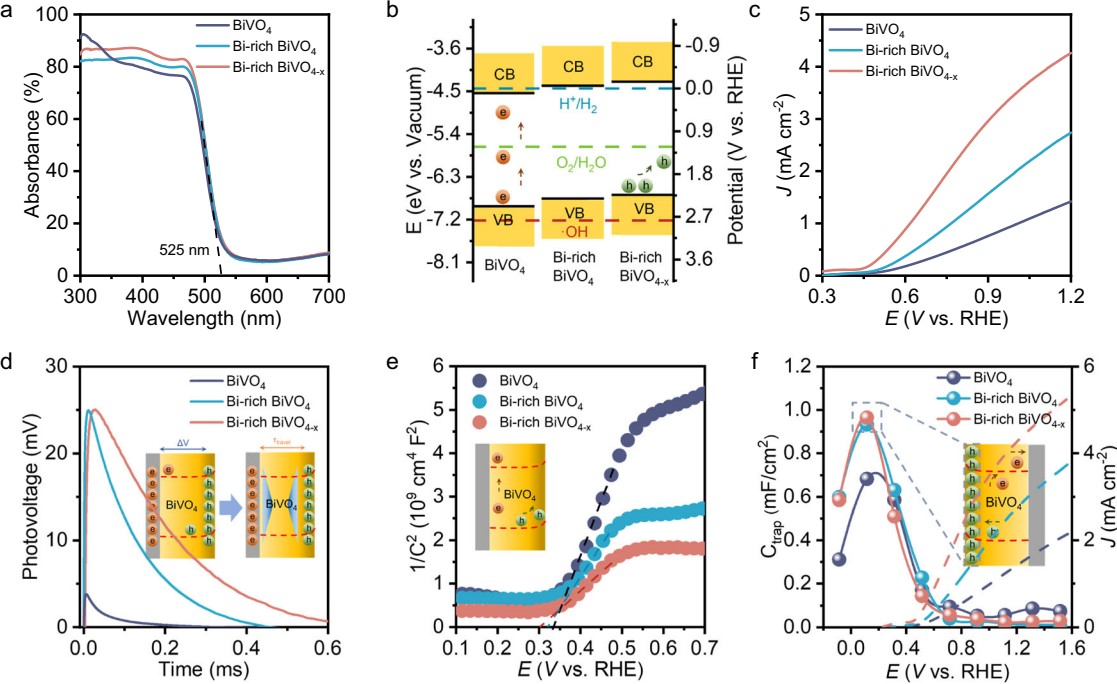

**Fig. 2 | PEC GOR performance and charge behavior characterization. a** UV–vis absorption spectra of $BiVO_4$, Bi-rich $BiVO_4$ and Bi-rich $BiVO_{4-x}$ photoanodes. **b** The bond alignments of $BiVO_4$, Bi-rich $BiVO_4$ and Bi-rich $BiVO_{4-x}$ photoanodes. **c** *J-V* curves of $BiVO_4$, Bi-rich $BiVO_4$ and Bi-rich $BiVO_{4-x}$ photoanodes in 0.5 M $Na_2SO_4$ (pH = 2) with 0.1 M glycerol under AM 1.5 G illumination. **d** The TS-SPV responses of $BiVO_4$, Bi-rich $BiVO_4$ and Bi-rich $BiVO_{4-x}$ photoanodes, and (**e**) Mott−Schottky plots measured under AM 1.5 G illumination conditions. **f** *J-V* curves (dashed lines) and $C_{trap}$ values (solid lines and dots) obtained on $BiVO_4$, Bi-rich $BiVO_4$ and Bi-rich $BiVO_{4-x}$ photoanodes.

V-$L_{2,3}$ peak shift in the region about 10 nm deep for Bi-rich $BiVO_4$ and Bi-rich $BiVO_{4-x}$ can be ascribed to V atoms loss. However, comparison of O-*K* edge of Bi-rich $BiVO_4$ and Bi-rich $BiVO_{4-x}$, the $O_v$ in Bi-rich $BiVO_{4-x}$ leads to a more obvious low-energy shift, which is consistent with the previous report[27]. X-ray photoelectron spectroscopy (XPS) spectra further provide reasonable evidence for the absence of V atoms and the formation of $O_v$ (Supplementary Fig. 12), where the blue shift of the V 2p peak and the increase of the $O_v$ peak are lined with the above.

UV-vis absorption spectra of $BiVO_4$, Bi-rich $BiVO_4$ and Bi-rich $BiVO_{4-x}$ show a similar absorption edge at 525 nm, whereas both Bi-rich surface and $O_v$ improve the light absorption ability to some extent (Fig. 2a). It is noting that the UV-vis absorption curve of bare $BiVO_4$ display two clear shoulders, which are ascribed to the charge transfer transition involving the V-O component and Bi and V centers[28]. Remarkably, the first shoulder (around 300–350 nm) of the Bi-rich $BiVO_4$ and Bi-rich $BiVO_{4-x}$ almost disappears, probably, because the charge-transfer transition centered at V is weakened due to the construction of the Bi-rich surface[23]. The band positions of all photoanodes are established by their Tauc plots and valence band (VB)-XPS (Supplementary Figs. 13, 14), and the band structure diagrams are displayed in Fig. 2b. Both the Bi-rich $BiVO_4$ and Bi-rich $BiVO_{4-x}$ present a slight shift toward the vacuum level relative to the $BiVO_4$. Under the premise that the VB position is appropriate, the upward shift of the band level is generally beneficial to achieve a more favorable band bending at the solid/liquid interface for efficient electron–hole separation[29,30].

Figure 2c and Supplementary Figs. 15–18 show the linear sweep voltammetry (LSV) curves of $BiVO_4$, Bi-rich $BiVO_4$, Bi-rich $BiVO_{4-x}$ photoanodes at a scanning rate of 20 mV s$^{-1}$ in a 0.5 M $Na_2SO_4$ electrolyte (pH = 2) with 0.1 M glycerol under AM 1.5 G illumination (100 mW cm$^{-2}$). The GOR photocurrent densities of the $BiVO_4$, Bi-rich $BiVO_4$, and Bi-rich $BiVO_{4-x}$ are 1.42, 2.74, and 4.26 mA/cm$^2$ at 1.23 V vs. RHE, respectively, indicating that both Bi-rich surface and $O_v$ can boost

the PEC oxidation performance of $BiVO_4$. To understand the role of Bi-rich surface and $O_v$ in PEC GOR, the charge transport efficiencies and the charge transfer efficiencies were evaluated by measuring their photocurrent densities using a hole scavenger (Supplementary Fig. 19) and calculating the theoretical photocurrent densities (Supplementary Fig. 20)[31,32]. It can be seen that the charge transport efficiency of bare $BiVO_4$ is 29.79%, which is increased to 53.48% after the formation of a Bi-rich surface, and further increased to 67.15% by the introduction of $O_v$ (Supplementary Fig. 21). Interestingly, for the carrier transfer efficiencies, the effect of Bi-rich surface is negligible, while the presence of $O_v$ leads to a substantial increase from the initial 81.97% to 98.86%.

Transient-state surface photovoltage (TS-SPV) response measurements (Fig. 2d) were employed to investigate the photogenerated charge dynamics process. In terms of SPV response, Bi-rich $BiVO_4$, and Bi-rich $BiVO_{4-x}$ exhibit stronger positive signals compared to $BiVO_4$, implying more favorable accumulation of photogenerated holes on the surface. Moreover, compared with the Bi-rich $BiVO_4$ photoanode, the Bi-rich $BiVO_{4-x}$ favor long-lived holes, which is benefiting for anodic oxidation[33,34]. Kelvin probe force microscopy (KPFM) further demonstrates the different surface potential caused by the Bi-rich surface and $O_v$ (Supplementary Figs. 22–24). Notably, the Bi-rich surface exhibits a noticeably brighter appearance compared to the pure counterparts (either $BiVO_4$ or $BiVO_{4-x}$). The increased brightness indicates a stronger charge separation-associated high surface potential, aligning with the findings from the TPV results[35]. The varied frequency Mott-Schottky (MS) curves for various photoanodes were analyzed to estimate reliable band edges. The band edge positions derived from these curves are consistent across different frequencies, indicating their frequency independence (Supplementary Figs. 25, 26). For a clearer comparison, Fig. 2e displays the MS curves of $BiVO_4$, Bi-rich $BiVO_4$, and Bi-rich $BiVO_{4-x}$ measured at 4000 Hz. As a result, both Bi-rich $BiVO_4$ and Bi-rich $BiVO_{4-x}$ exhibit a lower slope compared to $BiVO_4$, suggesting a higher donor density. In addition, the onset of MS plots for the Bi-rich $BiVO_4$ and Bi-rich $BiVO_{4-x}$ photoanodes demonstrated a gradual

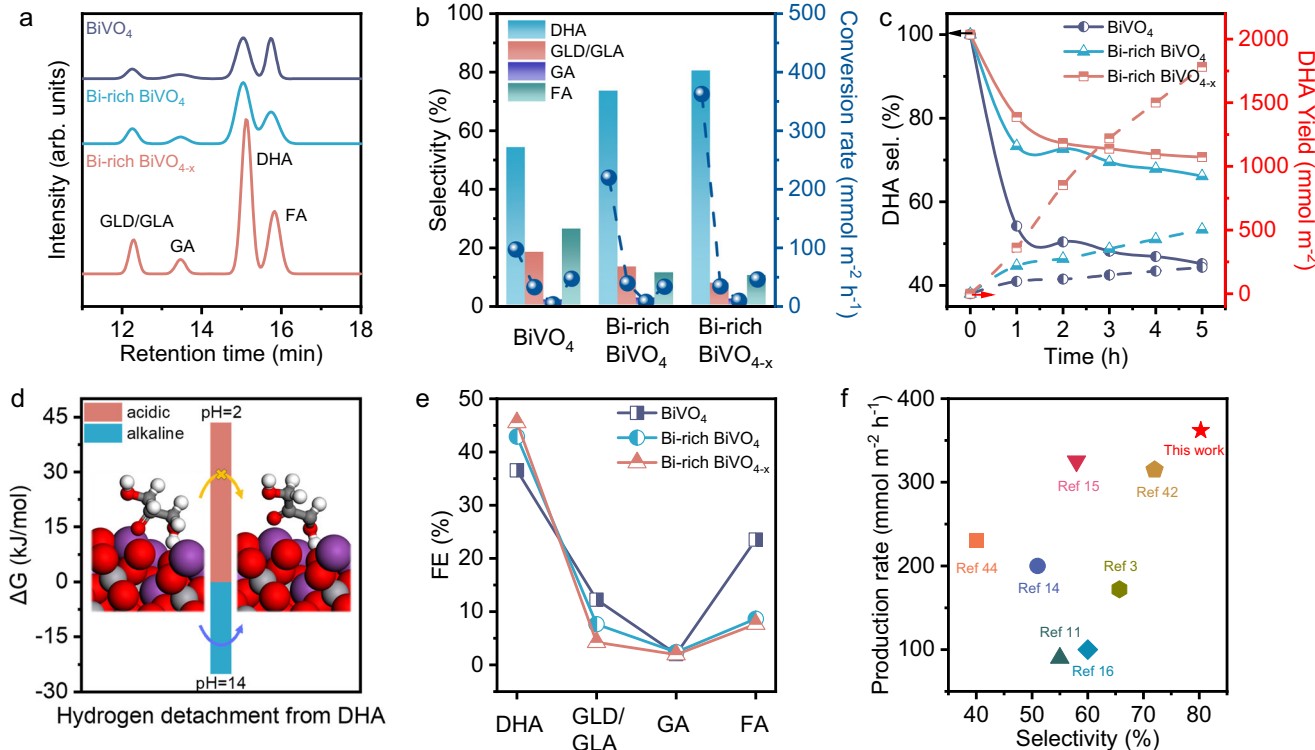

**Fig. 3 | Analysis of the product selectivity and conversion rate in PEC GOR.**
**a** HPLC spectra of the products and (**b**) conversion rate and selectivity of the main liquid products using the BiVO$_4$, Bi-rich BiVO$_4$, and Bi-rich BiVO$_{4-x}$ photoanodes under AM 1.5 G illumination at 1.23 V vs. RHE. **c** DHA selectivity and yields vs. reaction time under AM 1.5 G illumination at 1.23 V vs. RHE. **d** The change in the DFT-calculated Gibbs free energy associated with the initial hydrogen detachment for the decomposition of the DHA adsorbed on the Bi-rich BiVO$_{4-x}$ surface at two different pH conditions: pH = 2 (acidic) and pH = 14 (alkaline). **e** Faradaic efficiency of different products on BiVO$_4$, Bi-rich BiVO$_4$ and Bi-rich BiVO$_{4-x}$ photoanodes. **f** Summary of the DHA selectivity and conversion rate of PEC GLY oxidation by various photoanodes published in recent years.

cathodic shift, suggesting a greater band bending[36,37]. The cathodic shift of flat band potential is associated with the improved surface charge separation ability, which could imply the accumulation of photogenerated holes within the surface capacitive layer[38,39]. The photocurrent density and fitted capacitance of surface states (C$_{trap}$) calculated to prove this conclusion (Supplementary Fig. 27). As shown in Fig. 2f, the C$_{trap}$ values for Bi-rich BiVO$_4$ and Bi-rich BiVO$_{4-x}$ photoanodes are larger than that for BiVO$_4$ photoanode at the applied bias smaller than onset potential, which further indicates the holes accumulation in the surface capacitance layer. When the applied bias exceeds onset potential, the C$_{trap}$ values of Bi-rich BiVO$_4$ and Bi-rich BiVO$_{4-x}$ photoanodes are faster decreased than BiVO$_4$ photoanode, which can be ascribed to that a large number of accumulated charges are released rapidly and participate in the glycerol oxidation reaction, resulting in higher photocurrent density[40,41]. Overall, the Bi-rich surface creates a capacitive layer that is more conducive to the accumulation of photogenerated holes and stores sufficient charges for the subsequent oxidation reaction, which explains the better performance of the photoanodes of Bi-rich BiVO$_{4-x}$.

The products from PEC GOR were quantitatively analyzed by high-performance liquid chromatography (HPLC). Similar to previous reports[14–16], the main products of GOR using BiVO$_4$ as photoanode include dihydroxyacetone (DHA), glyceric acid (GLA), glyceraldehyde (GLD), glycolic acid (GA) and FA (Fig. 3a, Supplementary Fig. 28). The amount of glycerol can be determined by Nuclear Magnetic Resonance (NMR) analysis (Supplementary Fig. 29). The incorporation of the Bi-rich surface notably enhances the peak height of DHA, suggesting that the DHA emerges as the predominant product. All peak signals are witnessed further enhancement upon the introduction of O$_v$, indicating an improved conversion of GOR. The Bi-rich level and oxygen

vacancy concentration on the surface of BiVO$_4$ photoanodes is adjusted through control of their alkali soaking time and electroreduction duration to find the optimal conditions for DHA production (Supplementary Figs. 30, 31). The Bi-rich BiVO$_4$ and Bi-rich BiVO$_{4-x}$ photoanodes constructed under these conditions serve as the subjects of study in the following sections. The specific selectivity and conversion after one-hour reaction for BiVO$_4$, Bi-rich BiVO$_4$, BiVO$_{4-x}$ and Bi-rich BiVO$_{4-x}$ photoanodes are shown in Fig. 3b. It can be seen that the selectivity of DHA product is increased from 54.0% of BiVO$_4$ to 73.3% of Bi-rich BiVO$_4$, then to 80.3% of Bi-rich BiVO$_{4-x}$. The further enhanced selectivity of DHA for Bi-rich BiVO$_{4-x}$ might be originating from more Bi atom exposure due to the formation of O$_v$ (Supplementary Fig. 32). Correspondingly, the DHA production rate is elevated from 96.8 mmol m$^{-2}$ h$^{-1}$ of BiVO$_4$ to 219.2 mmol m$^{-2}$ h$^{-1}$ of Bi-rich BiVO$_4$, then to 361.9 mmol m$^{-2}$ h$^{-1}$ of Bi-rich BiVO$_{4-x}$. To understand the stability of GOR, the reaction time was extended to 5 h (Supplementary Fig. 33), and the selectivity and yield of the DHA are shown in Fig. 3c. As the reaction time increases, the selectivity of all photoanodes towards DHA experiences a certain degree of fading due to the occurrence of peroxidation reactions[42]. Finally, the selectivity of the Bi-rich BiVO$_{4-x}$ photoanode towards DHA tends to be stabilized after 5 h of reaction time, consistently maintaining a commendable performance level of 70.7%. Etching XPS was utilized to analyze the changes in the surface elemental concentration distribution of Bi-rich BiVO$_{4-x}$ photoanodes before and after 5 h GOR (Supplementary Fig. 34). As shown in Supplementary Fig. 35, the photoanode still maintained a surface Bi-rich state and abundant in O$_v$ after GOR, demonstrating its structural stability during prolonged working.

To explore the reason for the relatively stable accumulation of DHA in the reaction system, spin-polarized density functional theory

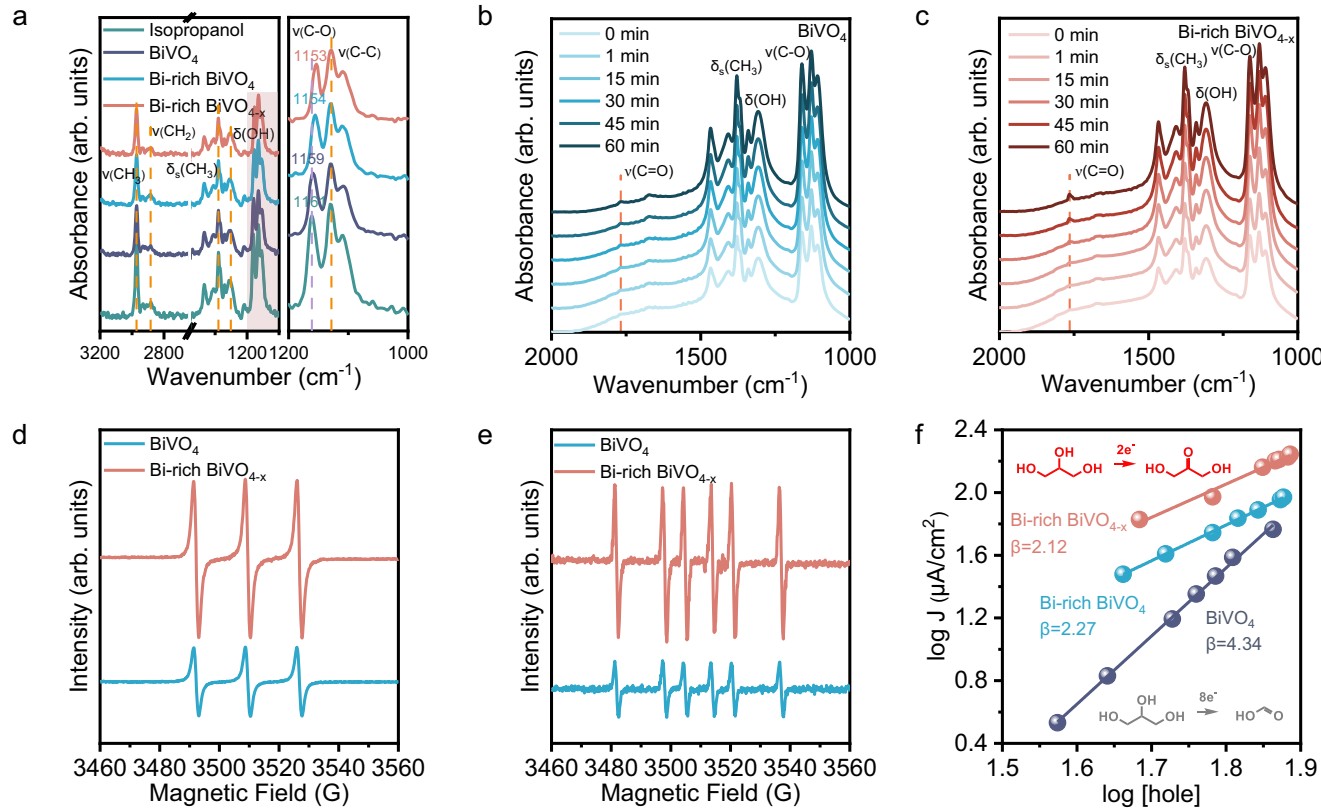

**Fig. 4 | Verification of glycerol adsorption on the photoanode surface and investigation into the glycerol oxidation mechanism. a** FT-IR spectra of isopropanol on $BiVO_4$, Bi-rich $BiVO_4$ and Bi-rich $BiVO_{4-x}$ photoanodes. **b, c** In situ FT-IR spectra of the dynamic oxidation process of isopropanol on $BiVO_4$, Bi-rich $BiVO_4$, and Bi-rich $BiVO_{4-x}$ photoanodes under AM 1.5 G illumination for 60 min. **d, e** EPR detection of photogenerated holes and carbon-centered radicals over illuminated $BiVO_4$ and Bi-rich $BiVO_{4-x}$ photoanodes. **f** Relationship between the photocurrent density and surface-hole density.

(DFT) was employed to calculate the desorption preference of DHA at the active sites (Supplementary Fig. 36 and Supplementary Table 1). The calculated value is negative (− 17.49 kJ/mol), indicating that DHA is not inclined to remain continuously adsorbed on the active sites for further overoxidation during the reaction process (Supplementary Fig. 37). The oxidation selectivity of glycerol substrate and DHA product is further demonstrated by PEC glycerol/DHA mixture oxidation, where the molar ratios of glycerol/DHA mixtures are established to be 10:1, 1:1 and 1:10, respectively (Supplementary Fig. 38). It can be seen that the photocurrent density of PEC glycerol/DHA mixture oxidation appears a drop compared to the photocurrent density of PEC GOR only when the molar ratio of glycerol/DHA mixture is reduced to 1:10. Further analysis of products reveals that the amount of DHA produced is almost same in the glycerol/DHA mixtures (Supplementary Figs. 39, 40). Therefore, the dropped photocurrent density in the low molar ratio of glycerol/DHA mixture might be ascribed to the factor of glycerol mass transfer, not that DHA is oxidized. Moreover, because the atomic spacing between two Bi atoms is about 3.09 Å (Supplementary Fig. 8), nearly twice that of a C-C single bond (1.54 Å). The spacing makes it difficult for neighboring carbon atoms to be adsorbed simultaneously, which may be a cause of the suppression of C-C bond breaking[43]. The above two reasons would act as main factor for continuous DHA production.

In addition, the effect of acidic and alkaline environments on its preference for carbon chain dehydrogenation (a key step in C-C bond breaking) was also noted. The calculation results indicate that the Gibbs free energy change for DHA dehydrogenation is 43.44 kJ/mol in acidic conditions, a positive value, while in alkaline conditions, it is − 25.06 kJ/mol, a negative value (Fig. 3d and Supplementary Table 2). This demonstrates that the carbon chain dehydrogenation process of

DHA is thermodynamically unfavorable under acidic conditions, which is consistent with the experimental results (Supplementary Fig. 41). Therefore, the low pH electrolyte we use similarly contributes to the stabilization of the carbon chain of DHA. The product selectivity and yield of Bi-rich $BiVO_{4-x}$ photoanode at different potentials are shown in Supplementary Fig. 42. Obviously, the product selectivity at different potentials is stable, but the conversion rate obviously increases with the increase of potential. This improvement is attributed to the larger photocurrent density at higher potentials. The Faradaic efficiency (FE) of each photoanode for different products are calculated and shown in Fig. 3e. The shortfall in the total FE was proven to be due to the production of oxygen (Supplementary Fig. 43). The FE of the products resulting from the primary hydroxyl oxidation reaction pathway (GLD/GLA, GA, FA) exhibited a decrease for the Bi-rich surface, while increasing the FE of DHA. The change in FE provides favorable evidence of a shift in the glycerol oxidation reaction pathway[44]. The PEC glycerol oxidative DHA performances in terms of selectivity and production rate are compared in Fig. 3f, demonstrating the best performances among all reported works.

To understand the role of Bi-rich surface in PEC GOR, in-situ Fourier transform infrared (FT-IR) spectra of $BiVO_4$, Bi-rich $BiVO_4$, and Bi-rich $BiVO_{4-x}$ photoanode are provided to investigate the changes in the adsorption state of glycerol. For comparison purposes, isopropanol and propanol with secondary and primary hydroxyl respectively, are employed as reagents. As shown in Fig. 4a, when isopropanol was adsorbed on the surfaces of $BiVO_4$, Bi-rich $BiVO_4$, and Bi-rich $BiVO_{4-x}$ photoanode, the peaks of $\nu$(C-O) band all shift to low wavenumbers. Notably, either $BiVO_4$ or $BiVO_{4-x}$ with Bi-rich surfaces exhibits more pronounced shifts (from 1161 to 1154 and 1153 cm$^{-1}$ respectively). The spectral alteration can be attributed to the bond

breaking caused by the adsorption of the hydroxyl group of iso-propanol to the surface exposed Bi atom, implying that Bi-rich surface is more conducive to the adsorption of secondary hydroxyl groups of glycerol[45,46]. In contrast, the adsorption of propanol did not cause any visible peak shift of each sample, which indicates that the primary hydroxyl groups of glycerol would be randomly oxidized on the surface of $BiVO_4$, regardless of surface exposed atoms (Supplementary Fig. 44). Furthermore, in situ FT-IR was performed to analyze the adsorption process of isopropanol on $BiVO_4$ and Bi-rich $BiVO_{4-x}$ photoanodes under AM 1.5 G illumination. As shown in Fig. 4b, c, a new characteristic peak is formed at 1765 $cm^{-1}$, attributing to the formation of the carbonyl group (C = O). The appearance of the carbonyl group implies the oxidation of isopropanol to acetone. The C = O signal detected on the surface of the Bi-rich $BiVO_{4-x}$ photoanode exhibits significantly greater intensity than that observed on the $BiVO_4$ pho-toanode over the course of time[42]. The more rapidly increasing signal intensity indicates that aldehyde products accumulate at a faster rate on the Bi-rich $BiVO_{4-x}$ photoanode surface, which is consistent with the superior DHA selectivity of Bi-rich $BiVO_{4-x}$ mentioned above. In addi-tion, the signal of the O = C-O bond cannot be observed in the FT-IR spectra of either $BiVO_4$ or Bi-rich $BiVO_{4-x}$ after one hour of illumination. The result was interpreted as the ketone products generated being able to remain in the system for an extended period, which corre-sponds to the DFT calculation results regarding DHA molecule deso-rption preference and carbon chain stability discussed above[3].

Room temperature electron spin resonance (ESR) spectroscopy was investigated to study the main intermediates in the PEC GOR process. As shown in Supplementary Fig. 45, in the absence of $H_2O_2$, production of ·OH by both the $BiVO_4$ and Bi-rich $BiVO_{4-x}$ pho-toanodes is negligible, which is consistent with the previous report[47]. Nevertheless, Bi-rich $BiVO_{4-x}$ exhibits a stronger photogenerated hole signal in comparison to $BiVO_4$, which is indicative of GOR occurring under the influence of photogenerated holes (Fig. 4d). While the signal intensity associated with carbon-centered radical is also enhanced to the same extent in the presence of glycerol (Fig. 4e)[48,49]. To further elucidate the reaction mechanism, the rate of glycerol oxidation by surface-trapped holes was determined by analyzing electrochemical impedance spectra (EIS) under varying light intensities (Supplemen-tary Fig. 46). The EIS spectra measured under different light intensities were fitted by equivalent model circuit and electrochemically active surface areas (Supplementary Figs. 47–49). The fitting results are shown in Supplementary Tables 3–5. The log/log plots of the photo-current density and hole density are displayed in Fig. 4f, and the reaction orders of $BiVO_4$, Bi-rich $BiVO_4$, and Bi-rich $BiVO_{4-x}$ photo-anodes can be established to be 4.34, 2.27, and 2.12 respectively. The different reaction orders suggest different reaction pathways. As the Bi-rich $BiVO_4$ and Bi-rich $BiVO_{4-x}$ photoanodes incline to oxidize the glycerol to DHA via a 2-electron transfer process (Supplementary Table 6), whereas the higher reaction order of $BiVO_4$ is due to the fact that its glycerol oxidation product contains more FA which is 8-electron transfer process[50]. The charge-transfer-related tendency exhibited by reaction order corresponds to the hole oxidation mechanism mentioned above[51].

To further validate the adsorption behavior of secondary hydro-xyl of glycerol on the Bi-rich surface, DFT calculations were conducted (Supplementary Fig. 50). The (010) facet of Bi-rich $BiVO_{4-x}$ served as the slab surface model for investigating adsorption energies and oxi-dation mechanisms of glycerol, the primary hydroxyl and secondary hydroxyl groups were individually adsorbed on the surface-exposed Bi atoms. As shown in Fig. 5a, the adsorption energy for the secondary hydroxyl group of glycerol is found to be − 97.33 kJ/mol, notably lower than − 67.39 kJ/mol predicted for the primary hydroxyl group of gly-cerol. This observation suggests a higher preference for the adsorption of the secondary hydroxyl group of glycerol on the surface of the Bi-rich $BiVO_{4-x}$ photoanode, making it thermodynamically susceptible

to oxidization to DHA (Fig. 5b)[52,53]. Furthermore, Fig. 5c shows the disparity in the adsorption energies of the secondary and primary hydroxyl groups of glycerol on the surfaces of pure $BiVO_4$ and Bi-rich $BiVO_{4-x}$, respectively. The adsorption capacity for primary hydroxyl groups of glycerol is observed to be even stronger than that for sec-ondary hydroxyl groups on the (010) facet of $BiVO_4$. Upon the removal of V and O atoms from the surface, there is a rapid increase in the adsorption strength for secondary hydroxyl groups. The observation further underscores the substantial impact of exposed Bi atoms on the adsorption behavior of the secondary hydroxyl group of glycerol. Finally, the Gibbs free energy profiles of the oxidation pathways for both primary and secondary hydroxyl groups of glycerol on Bi-rich $BiVO_{4-x}$ surface are shown in Fig. 5d. Evidently, Bi-rich $BiVO_{4-x}$ demonstrates thermodynamic favorability for each step in the oxida-tion pathway of secondary hydroxyl groups of glycerol compared to primary hydroxyl groups. The free energy difference of 61.9 kJ/mol throughout the entire reaction between the two pathways undeniably designates DHA as the more favorable reaction product.

## Discussion

In this work, a surface component tailoring approach was demon-strated to selectively oxidize glycerol to high value-added DHA. The Bi-rich and $O_v$ co-existed surface of $BiVO_4$ enabled the photocurrent density of GOR being improved from 1.42 to 4.26 mA $cm^{-2}$ at 1.23 V vs. RHE under AM 1.5 G illumination, while increasing the selectivity of DHA product from 54.0% to 80.3%, finally achieving a DHA production rate of 361.9 mmol $m^{-2}$ $h^{-1}$, marking the highest reported value to date. Comprehensive experimental detection and theoretical calculation confirm the strong electrostatic adsorption of glycerol secondary hydroxyl groups on the Bi-rich surface, bringing about a directional GOR pathway toward DHA via 2-electron transfer process. Meanwhile, the elevated surface potential engendered by the Bi-rich surface and the potent surface charge transfer facilitated by the oxygen vacancies provide favorable reaction dynamics for GOR. This work is expected to provide a scheme through surface atom tailoring instead of co-catalyst introduction to achieve a high-valued carbon chain-balanced product.

## Methods

### Synthesis of $BiVO_4$ seed layer on FTO substrates

The $BiVO_4$ seed layer was applied by spin-coating the precursor solu-tion onto pristine FTO substrates. To prepare the precursor solution for the $BiVO_4$ seed layer, 0.3234 g of Bi $(NO_3)_3·5H_2O$ (Sigma-Aldrich, purity > 99.99%) was dissolved in 1 ml of concentrated $HNO_3$ (PFP, 60 wt%), followed by the addition of 2 ml of Milli-Q water. Sub-sequently, 0.078 g of $NH_4VO_3$ (Sigma-Aldrich, purity ≥ 99%) and 0.167 g of polyvinyl alcohol (PVA, Sigma-Aldrich, purity ≥ 99%) were dissolved in the aforementioned solution and vigorously stirred until it achieved transparency. The precursor solution was then spin-coated onto the FTO substrate at 2500 rpm for 20 s, followed by calcination at 450 °C for 2 h in an air environment.

### Preparation of $BiVO_4$ photoanodes

0.1164 g of Bi $(NO_3)_3·5H_2O$ and 0.028 g of $NH_4VO_3$ were dissolved in 1.6 mL of concentrated $HNO_3$ (60 wt%). Milli-Q water was added until the total volume reached 60 mL. The $BiVO_4$ seed layer was immersed in the solution with the seed layer oriented downward. The solution was then transferred to a Teflon-lined autoclave and heated at 180 °C for a duration of 12 h. The resulting $BiVO_4$ was subsequently washed with Milli-Q water and subjected to calcination at 450 °C for 2 h in an air atmosphere.

### Preparation of Bi-rich $BiVO_4$ photoanodes

To prepare the Bi-rich $BiVO_4$ photoanode, start by dissolving 0.4 g of NaOH (Sigma-Aldrich, 97%) in 100 ml of deionized water. Next, extract 40 ml of the prepared solution and immerse the $BiVO_4$ photoanode in

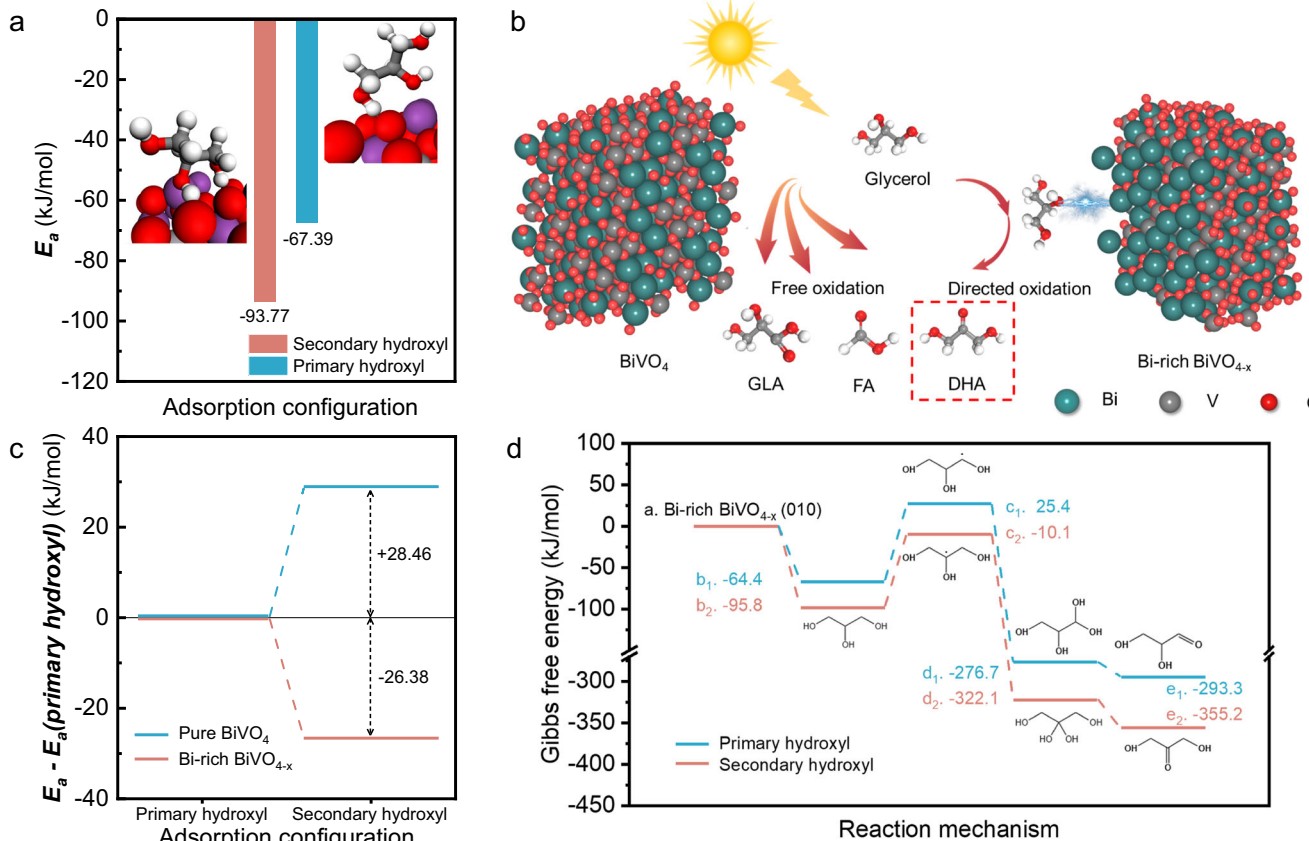

**Fig. 5 | Validation of experimental results through theoretical calculations.**
**a** The DFT-calculated energies related to glycerol adsorption on the Bi-rich BiVO$_{4-x}$ surface through either the primary or secondary hydroxyl group. **b** Schematic illustration of the PEC glycerol oxidation to DHA using Bi-rich BiVO$_{4-x}$ photoanode. **c** The adsorption energy involving secondary hydroxyl groups in relation to the adsorption energy associated with primary hydroxyl groups for glycerol adsorbed on both pure BiVO$_4$ and Bi-rich BiVO$_{4-x}$ surfaces. **d** The Gibbs free energy profiles linked to oxidation processes involving primary and secondary hydroxyl groups on Bi-rich BiVO$_{4-x}$ surfaces.

it for a period ranging from 60 to 200 s. This step is aimed at removing the surface V atoms. After the soaking process, carefully remove the photoanode, rinse it thoroughly with deionized water, and allow it to dry. The resulting Bi-rich BiVO$_4$ photoanode used in this study was obtained after soaking for 150 s.

### Preparation of Bi-rich BiVO$_{4-x}$ photoanodes

Oxygen vacancy generation was carried out in a three-electrode photoelectrochemical (PEC) cell, which included a 1 M potassium borate electrolyte (KBi) with a pH of 9.5. In this setup, a platinum sheet served as the counter electrode, while an Ag/AgCl electrode was employed as the reference electrode. The working electrode consisted of the Bi-rich BiVO$_4$ photoanode. It was maintained at a potential of $-0.8$ V vs. the Reversible Hydrogen Electrode (RHE) for a duration of 300 s to produce the Bi-rich BiVO$_{4-x}$ photoanode. BiVO$_{4-x}$ was also obtained using the same procedure.

### Material characterization

SEM images were captured using a JSM-7610F-Plus field emission scanning electron microscope. High-resolution transmission electron microscopy (HR-TEM) images and atom-level images were acquired using a JEOL JEM-ARM 200 F (NEOARM) transmission electron microscope with Cs-corrected/energy-dispersive X-ray spectroscopy (EDS)/ EELS capabilities. To determine the crystalline structures, X-ray diffraction (XRD) analysis was performed with a Siemens D500/5000 diffractometer using a Bragg–Brentano geometry, employing Cu Kα radiation at 40 keV and 40 mA. X-ray photoelectron spectroscopy (XPS) data were obtained using an SESXPS instrument (ESCA2000, VG

Microtech, England). UV-vis absorption spectra were recorded using a UV-vis spectrophotometer (Cary5000, Agilent) and obtained in diffuse reflection mode. Raman analyses were conducted using a Horriba Jovin Yvon LabRam Aramis Raman microscope equipped with a 532 nm laser. Inductively coupled plasma-mass spectrometry (ICP-MS) was carried out using an ICP-MS instrument from Agilent (model 7900). Kelvin probe force microscopy (KPFM) was conducted using an Atomic Force Microscope from Park Systems (model NX-10).

### PEC Measurements

The photoelectrochemical (PEC) performance of all photoanodes was evaluated using a three-electrode optical O-ring cell configuration. In this setup, a platinum (Pt) wire served as the counter electrode, and a saturated calomel electrode was used as the reference. For all PEC measurements, a 0.5 M sodium sulfate electrolyte (pH = 2, with or without 0.1 M glycerol) was consistently employed. Data acquisition was conducted using a CHI 660E electrochemical workstation. Illumination was provided by a solar simulator (100 mW/cm², Peccell Technologies, Yokohama, Japan, PEC-L01), with all electrodes being illuminated from the rear side. Linear sweep voltammetry (LSV) experiments were performed by sweeping the potential in the positive direction at a scan rate of 10 mV/s. The potential measured relative to the Ag/AgCl reference electrode was converted to the potential versus the Reversible Hydrogen Electrode (RHE) using the Nernst equation: E (vs. RHE) = E (vs. Ag/AgCl) + 0.0591 × pH + 0.196. Electrochemical impedance spectroscopy (EIS) measurements were taken by applying a sinusoidal AC perturbation of 5 mV across a frequency range from 0.1 Hz to 1 MHz. Mott-Schottky (MS) curves were obtained at

a frequency of 1000, 2000, 4000, and 8000 Hz with an amplitude of 10 mV[54].

During the evaluation of charge transfer efficiency ($\eta_{trans}$) and charge separation efficiency ($\eta_{sep}$), a $Na_2SO_3$ electrolyte was used as the hole scavenger. Here, $\eta_{sep}$ indicates the fraction of photogenerated holes at the electrode/electrolyte interface, whereas $\eta_{trans}$ denotes the fraction of these holes that reach the photoanode/electrolyte interface and participate in water oxidation.

The detailed calculation process of the absorbed photocurrent density in $BiVO_4$ films was as follows:

$$\eta_{trans} = \frac{J_{H_2O}}{J_{Na_2SO_3}} \tag{1}$$

$$\eta_{sep} = \frac{J_{Na_2SO_3}}{J_{abs}} \tag{2}$$

The single photon energy was calculated from Eq. (3):

$$E(\lambda) = h \times C/\lambda \tag{3}$$

where $E(\lambda)$ is the photon energy (J), h is the Planck constant ($6.626 \times 10^{-34}$ Js), C is the speed of light ($3 \times 10^8$ m s$^{-1}$) and $\lambda$ is the photon wavelength (nm).

The solar photon flux was then calculated according to Eq. (4):

$$Flux(\lambda) = P(\lambda)/E(\lambda) \tag{4}$$

where $Flux(\lambda)$ is the solar photon flux (m$^{-2}$ s$^{-1}$ nm$^{-1}$), and $P(\lambda)$ is the solar power flux (W m$^{-2}$ nm$^{-1}$). The theoretical absorbed photocurrent density under solar illumination (AM 1.5 G), $J_{abs}$ (A m$^{-2}$), was then calculated by integrating the solar photon flux between 300 and 525 nm, as shown in Eq. (6):

$$J_{abs} = e \times \int_{300}^{X} \eta_{har} Flux(\lambda) d\lambda \tag{5}$$

where e is the elementary charge ($1.602 \times 10^{-19}$ C), $\eta_{har}$ is the absorption spectrum[55].

## Photoelectrochemical glycerol oxidation measurements

PEC glycerol (GLY) oxidation measurements were conducted inside a sealed H-type glass cell over a 5-h duration. To separate the anode from the cathode chamber, a Nafion 212 proton exchange membrane was employed. The electrolyte solution, which included 0.1 M GLY, was composed of a 0.5 M $Na_2SO_4$ aqueous solution with the pH adjusted to 2 by adding a 0.5 M $H_2SO_4$ solution. All other experimental conditions were consistent with those utilized for the PEC water oxidation measurements.

To quantitatively analyze the glycerol oxidation products, PEC oxidation was carried out within a sealed H-type cell at a potential of 1.23 V vs. the Reversible Hydrogen Electrode (RHE) for a duration of 5 h. Following the reaction, 1 mL of the solution was withdrawn from the cell and subjected to analysis using high-performance liquid chromatography (HPLC), specifically an Agilent 1260 Infinity system. The HPLC system was equipped with an Aminex HPX-87 H column (Bio-Rad, 300 × 7.8 mm) was employed for analysis. The column was operated at a temperature of 50 °C and eluted with 10 mM aqueous $H_2SO_4$ as the eluent. The eluent was delivered at a flow rate of 0.5 mL/min. The detection wavelength of the DAD detector is 210 nm. The product selectivity and production rate of glycerol oxidation

reaction can be obtained from the following formula:

$$Selectivity(DHA) = \frac{n_{DHA}}{n_{all}} \times 100\% = \frac{n_{DHA}}{n_{DHA} + n_{GLA} + n_{GA} + n_{FA}} \times 100\%$$
$$= \frac{C_{DHA}}{C_{DHA} + C_{GLA} + C_{GA} + C_{FA}} \times 100\% \tag{6}$$

where $n_{GLA}$, $n_{DHA}$, $n_{GA}$ and $n_{FA}$ are the yields of DHA, GLA, GA and FA, respectively. C is the product concentration detected by HPLC. The selectivity of other liquid products was also calculated based on the above equation.

$$Production\ rate(DHA) = \frac{C_{DHA} \times V}{t \times A} \times 100\% \tag{7}$$

where V is the volume of the reaction solution, t is the reaction time, and A is the area of the photoanode. The production rate of other liquid products was also calculated based on the above equation.

$$Faradaic\ efficiency\ (DHA) = \frac{Number\ of\ holes\ to\ oxidize\ GLY\ to\ DHA}{Number\ of\ collected\ photogenerated\ holes} \times 100\%$$
$$= \frac{e_{DHA} \times n_{DHA} \times N}{Q/n} = \frac{2 \times C_{DHA} \times V \times N}{Q/n} \tag{8}$$

where $e_{DHA}$ is the number of holes required to oxidize one glycerol molecule to one DHA molecule, N is Avogadro's constant, Q is the quantity of electric charge, and n is the elementary charge. The Faradaic efficiency of other liquid products was also calculated based on the above equation ($e_{DHA} = 2$, $e_{GLA} = 4$, $e_{GA} = 5$, $e_{FA} = 8$).

$$Faradaic\ efficiency\ (GLA) = \frac{Number\ of\ holes\ to\ oxidize\ GLY\ to\ GLA}{Number\ of\ collected\ photogenerated\ holes} \times 100\%$$
$$= \frac{e_{GLA} \times n_{GLA} \times N}{Q/n} = \frac{4 \times C_{GLA} \times V \times N}{Q/n} \tag{9}$$

$$Faradaic\ efficiency\ (GA) = \frac{Number\ of\ holes\ to\ oxidize\ GLY\ to\ GA}{Number\ of\ collected\ photogenerated\ hole} \times 100\%$$
$$= \frac{\frac{2}{3} \times e_{GA} \times n_{GA} \times N}{Q/n} = \frac{\frac{2}{3} \times 5 \times C_{GA} \times V \times N}{Q/n} \tag{10}$$

$$Faradaic\ efficiency\ (FA) = \frac{Number\ of\ holes\ to\ oxidize\ GLY\ to\ FA}{Number\ of\ collected\ photogenerated\ holes} \times 100\%$$
$$= \frac{\frac{1}{3} \times e_{FA} \times n_{FA} \times N}{Q/n} = \frac{\frac{1}{3} \times 8 \times C_{FA} \times V \times N}{Q/n} \tag{11}$$

where $e_{GLA}$ is the number of holes required to oxidize one glycerol molecule to one GLA molecule, $e_{GA}$ is the number of holes required to oxidize one glycerol molecule to two-thirds GA molecule and $e_{FA}$ is the number of holes required to oxidize one glycerol molecule to three FA molecules[14].

## Calculation method of reaction order

The equivalent model circuit for fitting EIS results is shown in Figure S37. In this model, $C_{trap}$ represents the charges accumulated at surface states, $R_{trapping}$ represents the resistance in surface-hole trapping, and $R_{ct,trap}$ represents the resistance of interfacial charge transfer. The Nyquist plots exhibit two semicircles for this model. The high-frequency semicircle represents the process of hole trapping by surface states (hole accumulation at the surface), while the radius of the low-frequency semicircle reflects the process of interfacial hole transfer to $H_2O$.

The density of surface-trapped holes can be calculated by the following equation:

$$[\text{hole}] = C_{trap} \times V_{bias} \frac{R_{ct,trap}}{R_s + R_{trap} + R_{ct,trap}} / S \tag{12}$$

where $V_{bias}$ is the applied bias and S is the active area of the electrode.

The reaction rate was represented by the photocurrent density (*J*). The reaction order (*β*) of surface-trapped holes in water oxidation can be obtained by fitting the data.

$$J = k\,[\text{hole}]^{\beta} \tag{13}$$

$$\log J = \beta\,\log([\text{hole}]) + \log k \tag{14}$$

where k is the rate constant of the reaction, and β is the reaction order[56].

## Computational methods

All Density functional theory (DFT) calculations were performed using the Vienna ab initio simulation package[57,58]. The projector augmented wave method, along with the Perdew-Burke-Ernzerhof parameterization of the generalized gradient approximation with spin polarization, was employed for the exchange-correlation functional[59,60]. The plane-wave cutoff energy was additionally set to 520 eV. Hubbard U correction with *V* = 2.7 eV was applied to fully describe the strong *d*-electron correlation of transition metal, V[61]. During the geometry optimization of bulk phases, all atoms were fully relaxed with an energy convergence tolerance of $10^{-5}$ eV per atom and the final forces were converged to less than 0.02 eV/Å. Monkhorst-Pack grid points with $6 \times 3 \times 9$ *k*-meshes were utilized for optimizing bulk BiVO$_4$ phase[62]. The geometrically optimized bulk BiVO$_4$ model was cleaved into (010) direction, exposing surfaces with four-coordinated bismuth[24]. Each slab model had $2 \times 2 \times 1$ supercell to prevent interactions of guest molecules, glycerols, in adjacent cells, while keeping a vacuum thickness greater than 15 Å in the *z*-direction to prevent unrealistic interactions between neighboring images. Three layers of a total of nine atomic layers were allowed to have atomic relaxation under the same convergence tolerance as the bulk model. Monkhorst-Pack grid points with $4 \times 4 \times 1$ *k*-meshes were utilized for optimizing the slab models. The final bismuth-rich BiVO$_4$ (010) surface was developed through the optimization process. A variety of surface oxygen vacancy models were subsequently generated by removing an oxygen atom under the consideration of all potential configurations. Among these, the most stable bismuth-rich BiVO$_{4-x}$ surface model was finally obtained by the optimization of all these configurations.

The adsorption energy ($E_a$) of glycerol on a slab surface model can be defined as follows:

$$E_a = E_{s+glycerol} - \left(E_s + E_{glycerol}\right) \tag{15}$$

Here, $E_{s+glycerol}$, $E_s$, and $E_{glycerol}$ depict the DFT-calculated energies of a slab surface model with an adsorbed glycerol molecule, bare slab model, and glycerol molecule, respectively. Notably, a more negative adsorption energy indicates a stronger binding between the slab surface model and the glycerol molecule. The oxidation processes of glycerol on the bismuth-rich BiVO$_{4-x}$ surface model were further explored to describe the free energy profiles associated with the reaction. Notably, the enthalpic and entropic contributions to the free energy were incorporated by the vibrational frequency calculations. Intermediate species predicted during the oxidation processes were fully considered, referring to relevant literature[14].

## Data availability

Source data are provided with this paper.

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

## Acknowledgements

This work was supported by the NSFC (T2322013) and the Ministry of Science and ICT through the National Research Foundation of Korea (RS-2023-00302697). The TEM work at KENTECH was supported by the Center for Shared Research Facilities.

## Author contributions

K.Z. and J.H.P. conceived and designed the experiments. Y.L. and C.L. conducted the glycerol oxidation measurement and analysis. Y.L. carried out materials synthesis and electrochemical characterization. Y.L., J.M.M., Z.P.W., and S.H.O. carried out materials characterization. T.K.L. participated in part of the synthesis. B.G.L. and K.C.K. conducted the theoretical calculation. Y.L., K.Z., and J.H.P. co-wrote the paper. All authors discussed the results and commented on the manuscript. Y.L. and B.G.L. contributed equally to this work.

## Competing interests

The authors declare no competing interests.
