## [Peer Review File · Nature Communications]

Solar-Driven Highly Selective Conversion of Glycerol to Dihydroxyacetone Using Surface Atom Engineered BiVO₄ PhotoanodesREVIEWER COMMENTS

Reviewer #1 (Remarks to the Author):

It is quite exciting to learn from this manuscript that highly selective glycerol to DHA conversion could be achieved via a modified BiVO₄ photoanode. The selectivity toward the high value-added DHA is indeed very high. However, after finishing reading the manuscript, I do not think I learned much about the reason why Bi-rich BiVO_{4-x} photoanodes resulted in higher GOR selectivity toward DHA.

Firstly, it does not make sense that data in Fig. 3d provides direct support for the excellent stability exhibited by DHA in GOR as described in Line 203. Photocurrent for DHA, though lower than other substrates, is still significant. The significant photocurrent could mean that further oxidation and degradation of DHA to other products were very likely. It is hard to believe that DHA could accumulate over time without further conversion based on the data shown in Fig. 3d.

Secondly, it makes no sense to reach the argument that "... to oxidize the glycerol to DHA via a 2-electron transfer process whereas..." in Line 272 just based on the data in Fig. 4f. In fact, the reaction order for photogenerated holes could be 1 or 3 for water oxidation as shown in Ref. 50. The key is to identify the RDS. I would suggest the authors write out reaction equations in which glycerol converts to DHA and discuss in detail why the reaction order is 2. In addition, the range for hole concentrations is too narrow.

Thirdly, the DFT calculations only suggest that DHA as one of the GOR products is a likely pathway for Bi-rich BiVO_{4-x}. A more important question that is not answered in the manuscript is why activity toward C-C bond cleavage was not significantly observed.

I do not see any solid mechanistic evidence to support the observed high DHA selectivity. Therefore, I may not cite the high selectivity data reported in this manuscript in my future publications as I am not convinced the conclusions are fully supported. Considering Nature Communications is a high profile journal, I may not recommend the publication.

Other major issues for consideration and corrections:

1. M-S analysis should be conducted at variable frequencies to obtain reliable band edge estimation.
2. I believe ref. 40 is wrongly cited. So does the data point in Fig. 3f. Ref. 40 has nothing to do with GOR.
3. A more detailed and quantitative investigation of a series of Bi-rich BiVO_{4-x} materials is preferred for the elucidation of the superb GOR selectivity toward DHA.
4. Please specify whether UV-vis absorption spectra were acquired in diffuse reflectance mode or transmission mode. Be aware that absorbance does not have a unit so arbitrary unit is wrong. Please do not confuse absorption with adsorption in multiple places throughout the main text.

Reviewer #2 (Remarks to the Author):

The manuscript reports the use of BiVO₄ to perform photoelectrochemical (PEC) glycerol oxidation reaction (GOR), specifically with a target product of dihydroxyacetone (DHA). The use of BiVO₄ in PEC GOR has been reported increasingly in recent years, and the authors here modified the surface of BiVO₄ through an alkaline immersion method resulting in a Bi-rich surface with high oxygen vacancy concentration. This modification resulted in a significant increase of photocurrent (~3-fold) and perhaps more importantly, an increase of the DHA selectivity from 54 to 80%. The authors highlighted the DHA production rate that they achieved as outperforming other reported values in the literature. Overall, the study is interesting and relatively well written. However, there are several issues that need to be addressed before the manuscript can be re-considered for publication.

1. The authors claimed that there are negligible changes in morphologies between bare, Bi-rich and Bi-rich + oxygen vacancy BiVO₄. However, I believe the particle size shown on Fig. 1b is smaller than those on Fig. 1a and Supplementary Fig. 1. Unless, of course, if these images are not representatives; maybe some simple statistical quantification can be provided.
2. Supplementary Fig. 8: The authors need to identify clearer where is the surface and what is the difference between the left and right figure.
3. Line 116 - 118: The term UV-vis adsorption is incorrect. I believe it should be UV-vis absorption.
4. Line 155: I also believe that space charge density is an incorrect/inaccurate term. It should be donor density.
5. Related to the Mott-Schottky experiment. I understand that it was performed under single frequency. Have the authors confirmed that the Mott-Schottky assumptions (i.e., real impedance does not have any frequency dependence and imaginary impedance has a log-log frequency dependence with a slope of -1) are fulfilled within this frequency range (taking the plots at various frequencies within the range and showing that they do not change would be a good first indication)? If not, I am afraid the Mott-Schottky plot cannot be used to extract reliable data. Also, why can't the authors extract C_{sc} and plot the frequency-independent MS curve from the EIS data that they used for Fig. 2f?
6. Even if we assumed that the M-S plot is accurate, the extrapolation for the flatband (Fig. 1e) is not done properly as the x-axis is not placed at $y = 0$.
7. Fig. 3a and Supplementary Fig. 25: Where is the signal for glycerol? It is known that the signal for glycerol often overlaps with the GOR products, including DHA. Is this not the case for the authors experiment? If so, how did they ensure accurate quantification of glycerol vs. DHA?
8. Fig. 3e: The sum of the FE is not 100%. What would be the rest? Oxygen, I assume? Did the authors perform any oxygen detection measurements?

9. The authors wrote that their illumination source is a Xenon lamp equipped with AM1.5 filter and the intensity is 100 mW/cm². How was the calibration performed? It is important to provide the information since this step is often overlooked and performed incorrectly in the field. Also, this is especially important because the authors claimed a record performance.

Reviewer #3 (Remarks to the Author):

In this paper, the formation of DHA via the oxidation of glycerol by using BiVO₄ photoanode. The selective oxidation of secondary hydroxyl groups in glycerol is discussed based on the composition and band structure of the BiVO₄, especially in the surface region. It is claimed that catalysts with a Bi-rich surface show superior activity and selectivity for DHA formation. There are no major inconsistencies in the results presented. The following points require modification before publication.

1) Characterization of the composition and structure before and after the reaction should be performed, and the structural changes in the catalyst during the reaction should be discussed. In particular, changes in Bi concentration on the surface and changes in the concentration of oxygen vacancies should be shown.

2) For the following description of Fig. 4, more detailed explanation of the band structure is needed.

In addition, the signal of the O=C-O bond cannot be observed in the FT-IR spectra of either BiVO₄ or Bi-rich BiVO_{4-x} after one hour of illumination. The results can be considered that the energy band position of BiVO₄ is not easy to mineralize C=O species, which provides an explanation for the long-term retention of DHA in the PEC GOR3

3) In the oxidation reaction of glycerol, in addition to the adsorption of primary or secondary hydroxyl groups alone, adsorption involving both adjacent primary and secondary hydroxyl groups is often important. By comparing the reactivity of substrates with and without adjacent hydroxyl groups, it is possible to determine if adjacent hydroxyl groups affect your catalytic system. It is important to ascertain whether the influence of adjacent hydroxyl groups is negligible.

Response to the Comments on Nature Communications

Manuscript ID: NCOMMS-23-61325

Dear Reviewers:

We would like to thank the reviewers for careful reading and helpful comments. We revised the manuscript thoroughly according to the comments. The added items are highlighted in red in the main manuscript and the supplementary information. Following changes were made and listed below:

Reviewer(s)' Comments to Author:

Reviewer #1: It is quite exciting to learn from this manuscript that highly selective glycerol to DHA conversion could be achieved via a modified BiVO₄ photoanode. The selectivity toward the high value-added DHA is indeed very high. However, after finishing reading the manuscript, I do not think I learned much about the reason why Bi-rich BiVO_{4-x} photoanodes resulted in higher GOR selectivity toward DHA.

Our Response: We are very grateful for your pertinent comments on our work and for pointing out the shortcomings of our current manuscript. The questions and suggestions raised by you are extremely important and helpful, which make us deep thought, thereby improving the quality of our work in the revision. As will be shown below, we analyzed the reasons for the stable accumulation of DHA in the system from the perspectives of product desorption and carbon chain stabilization by supplementing the corresponding experiments and combining them with DFT calculations. The main mechanism of our work is further strengthened, and we believe the quality of the paper is significantly improved.

1#1: Firstly, it does not make sense that data in Fig. 3d provides direct support for the excellent stability exhibited by DHA in GOR as described in Line 203. Photocurrent for DHA, though lower than other substrates, is still significant. The significant photocurrent could mean that further oxidation and degradation of DHA to other products were very likely. It is hard to believe that DHA could accumulate over time without further conversion based on the data shown in Fig. 3d.

Our Response: Thanks for your nice question. Exploring the reasons behind the relatively stable accumulation of DHA in the system without rapid decomposition requires analysis from two perspectives: the desorption trend of DHA and the carbon chain equilibrium condition of DHA. First, we utilized DFT to calculate the energies of both the product (DHA) and the reactant (glycerol) when attached to the surface Bi active sites, and then subtracted them to determine the desorption preference of DHA. As shown on Fig. R1, the preference comes out as a negative value (-17.49 kJ/mol), which indicates that DHA is not prone to continuous adsorption on the active sites for further overoxidation during the reaction process. This is one of the reasons for the greater proportionate accumulation of DHA in the reaction system.

Fig. R1. The change in the DFT-calculated Gibbs free energy associated with the approach of the glycerol (new reactant) toward the DHA-adsorbed Bi-rich BiVO_{4-x} surface and subsequent release of the adsorbed DHA (product).

Second, it is also important to consider the balance of DHA carbon chains that remain in the active site. In previous works, DHA was considered to be more stable in low pH environments (Nat. Commun. 2019, 10, 1779, Nat. Commun. 2022, 13, 5848). Based on this, we directly used an electrolyte with a pH of 2 in this work, which may also contribute to the strongly continuous accumulation ability of DHA. To verify this hypothesis, we conducted GOR selectivity tests in electrolytes of different pH levels. The results revealed that as the pH value of the electrolyte increases, the selectivity for DHA rapidly decreases, being replaced by a higher selectivity for FA (Fig. R2). Obviously, the change of pH has a great influence on the stabilization of the C-C bond of DHA. While the dissociation of hydrogen on the carbon chain is a key step in C-C bond breakage (J. Phys. Chem. C. 2011, 115, 19702–19709, Top. Catal. 2012, 55, 280-289). Therefore, we further employed DFT calculations to simulate the carbon chain dehydrogenation preference of DHA in acidic and alkaline environments. The calculation results indicate that the Gibbs free energy change for DHA dehydrogenation is 43.44 kJ/mol in acidic conditions, a positive value, while in alkaline conditions, it is -25.06 kJ/mol, a negative value (Fig. R3). This demonstrates that the carbon chain dehydrogenation process of DHA is thermodynamically unfavourable under acidic conditions, which also provides evidence for the stable accumulation of DHA.

Fig. R2. Selective distribution of GOR products of Bi-rich BiVO_{4-x} photoanodes in electrolytes of different pH.

Fig. R3. The change in the DFT-calculated Gibbs free energy associated with the initial hydrogen detachment for the decomposition of the DHA adsorbed on the Bi-rich BiVO_{4-x} surface at two different pH conditions: pH=2 (acidic) and pH=14 (alkaline).

Above discussions have been added in our revised manuscript (Fig. 3d, Supplementary Fig. 35 and Supplementary Table 1-2).

1#2: Secondly, it makes no sense to reach the argument that “... to oxidize the glycerol to DHA via a 2-electron transfer process whereas...” in Line 272 just based on the data in Fig. 4f. In fact, the reaction order for photogenerated holes could be 1 or 3 for water oxidation as shown in Ref. 50. The key is to identify the RDS. I would suggest the authors write out reaction equations in which glycerol converts to DHA and discuss in detail why the reaction order is 2. In addition, the range for hole concentrations is too narrow.

Our Response: Thanks for your valuable recommendation. To elucidate why the photogenerated hole reaction order for the oxidation of glycerol to DHA is 2, we present the corresponding reaction equations as below:

The process primarily consists of two steps: the initial oxidation of glycerol to a radical intermediate (Eq. 1), and the further dehydrogenation and rearrangement of the radical intermediate to form DHA (Eq. 2). Both processes involve a one-electron transfer, making the complete oxidation of glycerol to DHA involve the transfer of two electrons. Furthermore, reaction orders of 1 or 3, as observed in Ref. 50, involve transformations between iron-oxygen species. However, such phenomena are difficult to observe in BiVO₄ photoanodes, which possess a generally stable valence state. Despite this, the water oxidation process may indeed involve a one-electron reaction that directly produces hydroxyl radicals. Yet, according to the data in Supplementary Fig. 41, the concentration of hydroxyl radicals is negligible. Therefore, a reaction order significantly lower than that for the oxygen evolution reaction can be attributed to the GOR process.

Secondly, the range of hole densities in our data, spanning 50 to 80 nm⁻², is broader than that in Ref. 50. The difference arises from our application of logarithmic scales to both the x-axis and y-axis data. This is because the reaction order and hole concentration follow the relationship outlined in Eq. 3, where J represents the steady-state photocurrent density, k is the apparent reaction rate constant, and β is the reaction order. To determine the specific value of the reaction order β, we take the logarithm of both sides of the equation. Consequently, β serves as the slope in Eq. 4 and can be directly obtained through linear fitting. The data presented in Ref. 50 features the steady-state OER rate (r_{OER}) as its vertical axis; thus, its horizontal axis can display the hole concentration without the need for logarithmic transformation. This accounts for the difference in our data presentation method compared to theirs.

$$J = k [\text{hole}]^\beta \quad (3)$$

$$\log J = \beta \log([\text{hole}]) + \log k \quad (4)$$

Above discussions have been added in our revised manuscript (Supplementary Eq. 1).

1#3: Thirdly, the DFT calculations only suggest that DHA as one of the GOR products is a likely pathway for Bi-rich BiVO_{4-x} . A more important question that is not answered in the manuscript is why activity toward C-C bond cleavage was not significantly observed.

Our Response: Thanks for your good advice. To explain why C-C bond breaking of DHA was not significantly observed, we supplemented the DFT calculations with both product desorption and C-C bond stabilization. Firstly, we calculated the energies of both the product (DHA) and the reactant (glycerol) when attached to the surface Bi active sites, and then subtracted them to determine the desorption preference of DHA. As a result, the preference comes out as a negative value (-17.49 kJ/mol), which indicates that DHA does not tend to continue adsorbing on the active sites (Fig. R1). This is one of the reasons it is not severely overoxidized. Secondly, we calculated the dehydrogenation preference of the carbon chain of DHA molecules in acidic and alkaline environments (Fig. R3). The results showed that the Gibbs free energy change for dehydrogenation in acidic environments is positive (43.44 kJ/mol), while in alkaline environments, it is negative (-25.06 kJ/mol). Since dehydrogenation is a key step in the C-C bond breakage of DHA, the acidic electrolyte we used is thermodynamically unfavourable for DHA bond cleavage. This is also a reason why DHA can accumulate to a high degree.

Above discussions have been added in our revised manuscript (Fig. 3d, Supplementary Fig. 35 and Supplementary Table 1-2).

1#4: M-S analysis should be conducted at variable frequencies to obtain reliable band edge estimation.

Our Response: Thanks for your valuable suggestion. To verify the reliability of the band edge estimations derived from the Mott-Schottky curves, they were reassessed at four distinct frequencies: 1000 Hz, 2000 Hz, 4000 Hz and 8000 Hz. As shown in Fig. R4, the onset of the Mott-Schottky plots for BiVO_4 , Bi-rich BiVO_4 , and Bi-rich BiVO_{4-x}

photoanodes exhibited no significant variation across different frequencies, indicating that their band edge positions remain largely unaffected by the applied frequency. To demonstrate the offset of the band edges more clearly, we continue to present the band edge positions at a consistent frequency for each sample within the main text of the manuscript. However, to avoid any misunderstanding among readers, we have included the following clarification in the manuscript.

“The varied frequency Mott-Schottky (MS) curves for various photoanodes were analyzed to estimate reliable band edges. The band edge positions derived from these curves are consistent across different frequencies, indicating their frequency independence (Supplementary Fig. 25-26). For a clearer comparison, Fig. 2e displays the MS curves of BiVO_4 , Bi-rich BiVO_4 , and Bi-rich BiVO_{4-x} measured at 4000 Hz. As a result, both Bi-rich BiVO_4 and Bi-rich BiVO_{4-x} exhibit a lower slope compared to BiVO_4 , suggesting a higher donor density. In addition, the onset of MS plots for the Bi-rich BiVO_4 and Bi-rich BiVO_{4-x} photoanodes demonstrated a gradual cathodic shift, suggesting a greater band bending³⁶⁻³⁷. The cathodic shift of flat band potential is associated to the improved surface charge separation ability, which could imply the accumulation of photogenerated holes within the surface capacitive layer³⁸⁻³⁹.”

Fig. R4. Varied frequency Mott-Schottky plots measured under AM 1.5 G illumination conditions for the BiVO_4 , Bi-rich BiVO_4 , and Bi-rich BiVO_{4-x} photoanodes.

Above discussions have been added in our revised manuscript (Fig. 3e, Supplementary Fig. 26).

1#5: I believe ref. 40 is wrongly cited. So does the data point in Fig. 3f. Ref. 40 has nothing to do with GOR.

Our Response: Thanks for your kind reminding. Upon scrutinizing the references in Fig. 3f, we found two labelling errors, Ref. 40 should be Ref. 42, and Ref. 41 should be Ref. 43. We have corrected the error there and put it back in Fig. 3f.

Fig. R5. Summary of the DHA selectivity and conversion rate of PEC GLY oxidation by various photoanodes published in recent years.

Above discussions have been added in our revised manuscript (Fig. 3f).

1#6: A more detailed and quantitative investigation of a series of Bi-rich BiVO_{4-x} materials is preferred for the elucidation of the superb GOR selectivity toward DHA.

Our Response: Thanks for your nice suggestion. To more precisely demonstrate the contributions of the Bi-rich surface and oxygen vacancies to the high selectivity for DHA, we explored these two factors separately. Initially, we controlled the Bi-rich level on the surface of pure BiVO₄ photoanodes by soaking them in alkali for varying durations (in 30 s intervals) and tested their impact on the selectivity for GOR. As shown on Fig. R6a, when the soaking time was only 30 and 60 s, the selectivity of the BiVO₄ photoanodes for DHA did not show a significant improvement, which could be due to the surface Bi atoms not being sufficiently exposed. However, when the duration was extended to 90 s, the selectivity for DHA rapidly increased, and after reaching 150 s, the selectivity essentially stabilized. This gradually slowing trend can be attributed to the soluble V atoms on the surface being essentially depleted by the alkali solution. When the soaking time reached

180 s, a downward trend in the LSV photocurrent began to emerge, making 150 s the optimal alkali soaking duration (Fig. R6b).

Figure R6. The a) GOR selectivity and b) LSV curves of Bi-rich BiVO_4 obtained at different alkali soaking times.

Subsequently, the contribution of oxygen vacancies to the selectivity for DHA was also studied. We conducted surface electroreduction on pure BiVO_4 photoanodes at 60 s intervals to construct oxygen vacancies at different concentrations. In contrast to the construction of a Bi-rich surface, while the formation of oxygen vacancies contributes to the selectivity for DHA, the duration of electroreduction does not appear to further influence its selectivity (Fig. R7a). The surface electroreduction process reduces some of the high-valence V and generates oxygen deficiencies, thereby also increasing the exposure of surface Bi atoms to a certain extent. However, this more intense method can modify the solid-liquid interface in a very short amount of time. Therefore, extending the treatment duration further does not have an additional impact on the selectivity for GOR. Based on this, it is sufficient to consider only the conditions that maximize photocurrent. Hence, we have determined 300 s to be the optimal duration for electroreduction treatment (Fig R7b). Finally, by integrating these two treatment approaches, we achieved an ideal DHA selectivity of 80.3% and an exceptional yield of $361.9 \text{ mmol m}^{-2} \text{ h}^{-1}$.

Fig. R7. The a) GOR selectivity and b) LSV curves of BiVO_{4-x} obtained at electrochemical reduction times.

Above discussions have been added in our revised manuscript (Supplementary Fig. 15 and (Supplementary Fig. 31).

1#7: Please specify whether UV-vis absorption spectra were acquired in diffuse reflectance mode or transmission mode. Be aware that absorbance does not have a unit so arbitrary unit is wrong. Please do not confuse absorption with adsorption in multiple places throughout the main text.

Our Response: Thanks for your kind reminding. The UV-Vis absorption spectra are obtained in diffuse reflection mode. In addition, we have corrected all unit and spelling errors, and we thank you for your careful reading.

Fig. R8. UV-vis absorption spectra of BiVO₄, Bi-rich BiVO₄ and Bi-rich BiVO_{4-x} photoanodes.

Above discussions have been added in our revised manuscript (Figure 2a and Methods).

Reviewer #2: The manuscript reports the use of BiVO₄ to perform photoelectrochemical (PEC) glycerol oxidation reaction (GOR), specifically with a target product of dihydroxyacetone (DHA). The use of BiVO₄ in PEC GOR has been reported increasingly in recent years, and the authors here modified the surface of BiVO₄ through an alkaline immersion method resulting in a Bi-rich surface with high oxygen vacancy concentration. This modification resulted in a significant increase of photocurrent (~3-fold) and perhaps more importantly, an increase of the DHA selectivity from 54 to 80%. The authors highlighted the DHA production rate that they achieved as outperforming other reported values in the literature. Overall, the study is interesting and relatively well written. However, there are several issues that need to be addressed before the manuscript can be re-considered for publication.

Our Response: We thank very much for your positive evaluation of our work. The questions and suggestions raised by you are extremely important and helpful, which inspired us to further optimize or extend our finding for broad interests. According to your suggestions, we have further examined and optimized many of the details in the manuscript and re-evaluated the results of debatable experiments such as M-S plots. We believe that the quality of our manuscript has been greatly improved with the help of your suggestions.

2#1: The authors claimed that there are negligible changes in morphologies between bare, Bi-rich and Bi-rich + oxygen vacancy BiVO₄. However, I believe the particle size shown on Fig. 1b is smaller than those on Fig. 1a and Supplementary Fig. 1. Unless, of course, if these images are not representatives; maybe some simple statistical quantification can be provided.

Our Response: Thanks for your nice question. The grain size differences in Fig. 1a-b and Supplementary Fig. 1 are the result of local differences in the magnification of the shooting scale. To avoid this confusion, we provide SEM images at smaller shot magnifications (Fig. R9a-c), and for each sample 100 grains were sampled for simple statistics on the distribution of sizes (Fig. R9d-f). As can be seen from the above results, the variation in the morphology of each sample is almost negligible.

Fig. R9. a-c) SEM images and d-f) particle size distribution of BiVO₄, Bi-rich BiVO₄, and Bi-rich BiVO_{4-x} photoanodes at lower magnifications.

Above discussions have been added in our revised manuscript (Supplementary Fig. 2).

2#2: Supplementary Fig. 8: The authors need to identify clearer where is the surface and what is the difference between the left and right figure.

Our Response: Thanks for your kind reminding. We have added the following note under Supplementary Fig. 8 to help readers better understand it.

As shown above, the surface of pure BiVO₄ tends to alternate between Bi and V atoms (left), a state in which the surface exposed Bi atoms are limited. By employing alkali leaching, the most surface-covered V atoms can be removed, leaving Bi-rich surface (right).

Fig. R10. The Schematic diagrams of original BiVO₄ surface (left) and Bi-rich surface (right).

Above discussions have been added in our revised manuscript (Supplementary Fig. 9).

2#3: Line 116 - 118: The term UV-vis adsorption is incorrect. I believe it should be UV-vis absorption.

Our Response: Thanks for your kind reminding. We have corrected each of these spelling errors.

2#4: Line 155: I also believe that space charge density is an incorrect/inaccurate term. It should be donor density.

Our Response: Thanks for your kind reminding. We have corrected it.

2#5: Related to the Mott-Schottky experiment. I understand that it was performed under single frequency. Have the authors confirmed that the Mott-Schottky assumptions (i.e., real impedance does not have any frequency dependence and imaginary impedance has a log-log frequency dependence with a slope of -1) are fulfilled within this frequency range (taking the plots at various frequencies within the range and showing that they do not change would be a good first indication)? If not, I am afraid the Mott-Schottky plot cannot be used to extract reliable data. Also, why can't the authors extract C_{sc} and plot the frequency-independent MS curve from the EIS data that they used for Fig. 2f?

Our Response: Thanks for your valuable suggestion. We first verified the applicable frequency range of the Mott-Schottky hypothesis by extracting Bode plots from the EIS data. As shown on Fig. R11, neither the real nor the imaginary parts of impedance satisfy the Mott-Schottky assumption at low frequencies (< 1000 Hz). This is attributed to the accumulation of photogenerated charges on the surface of photoanodes affecting the width and potential distribution of the space charge region. Such effects are particularly pronounced in the low-frequency range where charges cannot rapidly diffuse. Furthermore, the samples we constructed exhibit a surface state rich in defects, far from an ideal condition. Consequently, it is challenging to directly observe the trends articulated by the

Mott-Schottky assumption in the low-frequency region. To mitigate data inaccuracies arising from this issue, we retested the varied frequency Mott-Schottky plots within the 1000-8000 Hz range, which aligns with the Mott-Schottky assumption. The variation in the slope of the M-S curves at different frequencies can be attributed to the photogenerated electron-hole pairs formed under illumination, which exhibit different dynamic characteristics with changes in the frequency of the test signal. While the flat band potentials of BiVO_4 , Bi-rich BiVO_4 , and Bi-rich BiVO_{4-x} photoanodes, measured at various frequencies, showed no significant changes, demonstrating the stability of the space charge region formed at the interface between the photoanode and the electrolyte (Fig. R12). This indirectly confirms the stability of the electrochemical behaviour of the photoanode materials within this frequency range.

Fig. R11. Bode plots extracted from the EIS data for the BiVO_4 , Bi-rich BiVO_4 , and Bi-rich BiVO_{4-x} photoanodes.

Fig. R12. Varied frequency Mott-Schottky plots measured under AM 1.5 G illumination conditions for the BiVO_4 , Bi-rich BiVO_4 , and Bi-rich BiVO_{4-x} photoanodes.

In addition, we also tried to extract C_{sc} from EIS data to plot the frequency-independent MS curve according to your advice. Although this approach enabled us to obtain MS curves with essentially correct trends, sampling as densely as with direct testing of MS curves proved challenging (Fig. R13). Consequently, accurately determining the flat band potential was difficult, introducing a degree of subjectivity. To obtain a more objective edge cut, we performed curve fitting on the points within the -0.1-1.5 V range and extended the tangent line to the X-axis to determine the position of the flat band potential. Although our final results align with the trends observed in directly tested Mott-Schottky curves, to avoid misunderstanding among readers, we believe presenting the varied frequency MS curve results directly is more straightforward and comprehensible.

Fig. R13. The Mott-Schottky plots, independent of frequency, constructed from the space charge capacitance (C_{sc}) extracted from EIS measurements under various biases.

Above discussions have been added in our revised manuscript (Supplementary Fig. 25-26).

2#6: Even if we assumed that the M-S plot is accurate, the extrapolation for the flatband (Fig. 2e) is not done properly as the x-axis is not placed at $y = 0$.

Our Response: Thanks for your kind reminding. We have corrected it. In addition, according to the previous reply, we replaced the data in Fig. 2e with MS plots measured at 4000 Hz to make the presented results more reliable (Fig. R14).

Fig. R14. Mott–Schottky plots measured at 4000 Hz under AM 1.5 G illumination conditions.

Above discussions have been added in our revised manuscript (Fig. 2e).

2#7: Fig. 3a and Supplementary Fig. 25: Where is the signal for glycerol? It is known that the signal for glycerol often overlaps with the GOR products, including DHA. Is this not the case for the authors experiment? If so, how did they ensure accurate quantification of glycerol vs. DHA?

Our Response: Thanks for your nice question. To analyze GOR products, we utilized HPLC equipped with a DAD (Diode Array Detector) detector. The DAD detector is a type of ultraviolet detector, and since glycerol molecules do not possess the ability to absorb ultraviolet light, their peak will not appear in the HPLC chromatogram. Furthermore, although an RI (differential Refractive Index) detector can detect signal peaks of glycerol, as you mentioned, its signals may overlap with those of other products, making it not an optimal choice.

The quantification of glycerol was conducted using the internal standard method in NMR (Nuclear Magnetic Resonance). The position of glycerol in the NMR spectrum is essentially unaffected by solvent peaks, facilitating integration (Fig. R15). Additionally, chemically stable DMSO (Dimethyl sulfoxide) was used as the internal standard. The specific amount of glycerol was calculated by computing the ratio of the peak areas of glycerol to DMSO. This method allows us to accurately determine the ratio of glycerol to DHA.

Fig. R15. NMR spectra of glycerol oxidation products.

(This Figure is for reviewers only)

2#8: Fig. 3e: The sum of the FE is not 100%. What would be the rest? Oxygen, I assume?
 Did the authors perform any oxygen detection measurements?

Our Response: Thanks for your wise question. The oxidation product other than GOR is oxygen, we have performed the measurement using a high-performance gas chromatography system with sampling and analysis executed every half hour. As mentioned, in addition to hydrogen generated at the cathode, the gas-phase products also contain a certain amount of oxygen (Fig. R16a). Accordingly, the Faraday efficiency for oxygen generation was calculated, the total Faraday efficiency of GOR and OER is close to 100% (Fig. R16b).

Fig. R16. a) The actual quantities of H₂ and O₂ evolution in GOR under AM 1.5 illumination and b) total Faraday efficiency for all products.

Above discussions have been added in our revised manuscript (Supplementary Fig. 39).

2#9: The authors wrote that their illumination source is a Xenon lamp equipped with AM1.5 filter and the intensity is 100 mW/cm². How was the calibration performed? It is important to provide the information since this step is often overlooked and performed incorrectly in the field. Also, this is especially important because the authors claimed a record performance.

Our Response: Thanks for your valuable question. Our inaccurate descriptions may lead to your misunderstanding. In this work, the light source is solar simulator (Peccell Technologies). As shown in the figure below, we used a radiometer to display the real-time light intensity, and the probe of the radiometer was placed at the platform marker. Calibration is done by moving the platform back and forth/changing the energy of the Xenon lamp until the readout of the radiometer shows the intensity of 1 standard sunlight. For PEC testing, we use a quartz electrolytic cell that absorbs almost no light and keep the position of the photoanode at the platform mark (where the radiometer probe is placed). Based on this, we can ensure a light intensity of 100 mW/cm² on the photoanode.

Fig. R17. Xenon lamp intensity calibration by radiometer.

(This Figure is for reviewers only)

Reviewer #3: In this paper, the formation of DHA via the oxidation of glycerol by using BiVO₄ photoanode. The selective oxidation of secondary hydroxyl groups in glycerol is discussed based on the composition and band structure of the BiVO₄, especially in the surface region. It is claimed that catalysts with a Bi-rich surface show superior activity and selectivity for DHA formation. There are no major inconsistencies in the results presented. The following points require modification before publication.

Our Response: We thank very much for your positive evaluation of our work. The questions and suggestions raised by you are of great importance and assistance. Based on these, we have given serious thought and made efforts to enhance the quality of our work. As will be shown below, we analyzed the changes in surface element concentrations before and after the reaction of the samples and evaluated the effect of adjacent hydroxyl groups on the final product of the reaction. The main contribution of our work is further strengthened, and we believe the quality of the paper is significantly improved.

3#1: Characterization of the composition and structure before and after the reaction should be performed, and the structural changes in the catalyst during the reaction should be discussed. In particular, changes in Bi concentration on the surface and changes in the concentration of oxygen vacancies should be shown.

Our Response: Thanks for your nice suggestion. The change in surface element concentration of the photoanode before and after the reaction is an important measure of its stability. In order to visualize the changes in surface elemental concentrations, we performed etching-XPS tests on Bi-rich BiVO_{4-x} photoanodes before GOR and after 5 h GOR, respectively (Fig. R18). The XPS spectra of the two photoanodes were collected every 40 s of etching to obtain the change in elemental concentration from the surface to the bulk phase. By analyzing the acquired XPS spectra, the percentage of elements at different depths can be obtained, which is organized and presented in Fig. R19. As shown in Fig. R19, the elemental percentage of O rises rapidly after a single etching and stabilizes in the region after that, which is caused by the surface-constructed O vacancies. The almost identical variation of the elemental percentage of O in the two figures proves that the O vacancies are in a relatively stable state in the GOR. In addition, although the atomic percentage of element V is higher than that of element Bi in the bulk phase, the percentage

of element Bi is clearly higher than that of element V at the surface. This is caused by the conformation of the Bi-rich surface. After 5 h of GOR, the elemental distribution of the Bi-rich BiVO_{4-x} photoanode did not change significantly, and the apparent Bi-rich state was still maintained at the surface, which demonstrated the surface stability of the photoanode in GOR.

Fig. R18. Etching XPS spectra of Bi-rich BiVO_{4-x} photoanodes before GOR and after 5 h GOR. Etching was performed every 40 s for a total of 10 etchings. Under the effect of argon ion etching, part of Bi^{3+} was reduced to metallic Bi, so that the peaks of metallic Bi appeared at about 157.2 eV and 162.5 eV in the XPS spectra of Bi 4f.

Fig. R19. Summary of elemental percentages of Bi-rich BiVO_{4-x} photoanodes before GOR and after 5 h GOR obtained from etching-XPS spectra. The element C present in this result is mainly from carbon contamination during XPS testing.

Above discussions have been added in our revised manuscript (Supplementary Fig. 33-34).

3#2: For the following description of Fig. 4, more detailed explanation of the band structure is needed.

In addition, the signal of the O=C-O bond cannot be observed in the FT-IR spectra of either BiVO_4 or Bi-rich BiVO_{4-x} after one hour of illumination. The results can be considered that the energy band position of BiVO_4 is not easy to mineralize C=O species, which provides an explanation for the long-term retention of DHA in the PEC GOR³

Our Response: Thanks for your kind reminding. After careful consideration, we believe that the description of the band positions here is inappropriate. Although we have provided specific band structures of the photoanodes above (Fig. 2b), accurately describing the specific band positions for the mineralization of C=O species proves challenging. In the revised manuscript, we have included additional calculations concerning DHA desorption and carbon chain equilibrium to explain the stability of its molecules. Therefore, we have updated the description in this section to the following content, aiming to enhance the rigor of the logic of manuscript.

“In addition, the signal of the O=C-O bond cannot be observed in the FT-IR spectra of either BiVO_4 or Bi-rich BiVO_{4-x} after one hour of illumination. The result was interpreted as the ketone products generated being able to remain in the system for an extended period, which corresponds to the DFT calculation results regarding DHA molecule desorption preference and carbon chain stability discussed above.”

3#3: In the oxidation reaction of glycerol, in addition to the adsorption of primary or secondary hydroxyl groups alone, adsorption involving both adjacent primary and secondary hydroxyl groups is often important. By comparing the reactivity of substrates with and without adjacent hydroxyl groups, it is possible to determine if adjacent hydroxyl groups affect your catalytic system. It is important to ascertain whether the influence of adjacent hydroxyl groups is negligible.

Our Response: Thanks for your nice suggestion. To explore whether adjacent hydroxyl groups affect the reaction pathway of the oxidation process, we utilized 1,2-Propanediol (simulating the presence of adjacent hydroxyl groups) and 1,3-Propanediol (simulating the absence of adjacent hydroxyl groups) as reaction substrates and examined their oxidation products. After subjecting the post-reaction solutions to Nuclear Magnetic Resonance Spectroscopy (NMR) analysis, it was found that the main products were the result of single hydroxyl groups being oxidized (Fig. R20). Products resulting from the simultaneous oxidation of two hydroxyl groups were not observed. This implies that the active sites on Bi-rich BiVO_{4-x} prefer the adsorption of single hydroxyl groups, indicating that even in the presence of adjacent hydroxyl groups, it is challenging for them to be adsorbed and oxidized simultaneously. Based on this observation, we believe that the influence of adjacent hydroxyl groups on the reaction can be considered negligible.

Fig. R20. NMR spectra of the products from the oxidation of 1,2-Propanediol and 1,3-Propanediol using Bi-rich BiVO_{4-x} photoanodes.

(This Figure is for reviewers only)

REVIEWER COMMENTS

Reviewer #1 (Remarks to the Author):

1. I do not think the author response answered my first concern about the original Fig. 3d that seems to be removed in this revision.
2. All of the hole density plots were log-log plots in Ref. 50 as they should be. The data in Fig. 4f only present $\sim 70\text{-}80\text{ nm}^{-2}$ for Bi-rich BiVO₄. Will it be possible to cover the range from 50 to 80 nm^{-2} ?
3. For my previous third question, I am sorry to say that results from DFT calculations alone to answer such an important question are not sufficient to impress experimental chemists. Computations are supposed to be supplementary tools in an experiment-oriented paper. Is there any experimental support for the question?

Reviewer #2 (Remarks to the Author):

The authors have satisfactorily addressed my comments to the initial version of manuscript. I have no objection to the publication of this revised manuscript, except for one suggestion: perhaps the authors can consider including the NMR analysis into the manuscript/Sl. I suppose this might be useful for some readers.

Reviewer #3 (Remarks to the Author):

The manuscript is well revised according to the comments. Therefore, now the manuscript is acceptable for publication.

Response to the Reviewers: Manuscript ID: NCOMMS-23-61325A

Dear Reviewers

We would like to thank the reviewers for careful reading and helpful comments. We revised the manuscript thoroughly according to the comments. The added items are highlighted in red in the main manuscript and the supplementary information. Following changes were made and listed below:

Reviewer(s)' Comments to Author:

Reviewer #1:

1#1: I do not think the author response answered my first concern about the original Fig. 3d that seems to be removed in this revision.

Our Response: Thank you for raising concerns once again regarding DHA oxidation. In our original Figure 3d, we presented a comparison of photocurrents across various reaction substrates, including GLD, GA, GLA, Glycerol, and DHA, among others. These results are intended to illustrate both the stability of the material and the challenges inherent in the reaction process. The stability primarily relates to the photoelectrode, while the difficulty of the reaction pertains to activity trends. Consequently, these findings may raise another issue: the oxidation potential of DHA. We acknowledge your point that DHA oxidation could affect its accumulation during GOR. To address your inquiries, in the first revised manuscript, we conducted DFT calculations comparing the adsorption and desorption energies of DHA and Glycerol at the Bi site (see Figure S35). The results clearly indicate a higher activity trend for Glycerol oxidation compared to DHA oxidation, suggesting a preference for Glycerol oxidation in the presence of a DHA and Glycerol mixture.

To further elucidate the activity trend, in this revised manuscript, we compared the photoelectrochemical (PEC) performances under varying ratios of Glycerol and DHA. Specifically, a 40 mL electrolyte initially containing 4 mmol Glycerol had different amounts of DHA added to it, with molar ratios of Glycerol to DHA set at 10:1, 1:1, and 1:10, respectively. As depicted in Figure R1, the photocurrent density notably declines with a 1:10 molar ratio of Glycerol to DHA. Subsequently, we subjected mixed solutions of DHA and Glycerol to GOR conditions for 5 hours and analyzed the resulting products. As demonstrated in Figure R2, DHA levels increased markedly in the 10:1 and 1:1 molar ratios of Glycerol to DHA, while the increase was minimal in the 1:10 ratio. This observation may be attributed to Glycerol's mass transfer factor rather than DHA oxidation.

Fig. R1. J - V curves of Bi-rich BiVO_{4-x} photoanode in mixed solutions with different proportions of glycerol and DHA.

Fig. R2. Comparison of HPLC spectra of Bi-rich BiVO_{4-x} photoanode before and after 5 hours of reaction in mixed solutions of glycerol and DHA in different proportions.

Overall, we have clarified the above results.

The above discussions have been added to our revised manuscript (Supplementary Fig. 38-40).

1#2: All of the hole density plots were log-log plots in Ref. 50 as they should be. The data in Fig. 4f only present ~ 70 - 80 nm^{-2} for Bi-rich BiVO_4 . Will it be possible to cover the range from 50 to 80 nm^{-2} ?

Our Response: Thanks for your good question. The variation of hole density in Fig. 4f is dominated by the variation in light intensity. So, it is possible to cover hole density in the range of 50 - 80 nm^{-2} as long as the light intensity is continuously reduced. To further expand the range of hole densities, we conducted additional tests on the original samples under

two different light intensities: 34.62 mW/cm² and 23.75 mW/cm². As shown in Figure R1, a noticeable decrease in hole density across all samples is observed when the light intensity is further reduced. The hole density coverage of BiVO₄, Bi-rich BiVO₄ and Bi-rich BiVO_{4-x} photoanodes in this range of light intensities is 37.50 to 72.95 nm⁻², 45.92 to 75.34 nm⁻² and 48.31 to 76.91 nm⁻², respectively. Furthermore, with the addition of extra test points, the reaction orders obtained through a linear fitting for BiVO₄, Bi-rich BiVO₄ and Bi-rich BiVO_{4-x} photoanodes have changed to 2.12, 2.27, and 4.37, respectively. We believe that these reaction orders, derived from a broader range of hole densities, will become more reliable.

Fig. R1. Relationship between the photocurrent density and surface-hole density.

The above discussions have been added to our revised manuscript (Fig. 4f, Supplementary Fig. 46 and Supplementary Table 3-5).

1#3: For my previous third question, I am sorry to say that results from DFT calculations alone to answer such an important question are not sufficient to impress experimental chemists. Computations are supposed to be supplementary tools in an experiment-oriented paper. Is there any experimental support for the question?

Our Response: Thanks for your concerns about C-C cleavage, and this is very nice question. According to previous report (J. Am. Chem. Soc. 2023, 145, 25382–25391), the cleavage of the C-C bond is the result of two neighboring carbons being extracted both electrons and protons. For Bi-rich BiVO₄, the atomic spacing between two Bi atoms is about 3.09 Å (Supplementary Fig. 8), nearly twice that of a C-C single bond (1.54 Å). This spacing makes it difficult for neighboring carbon atoms to be adsorbed simultaneously,

which may be a cause of the suppression of C-C bond breaking. However, for the common BiVO_4 , the C-C single bond is able to be adsorbed simultaneously at Bi and V atom, which therefore, leads to low selectivity of DHA.

Until now, there are no experimental tool that can observe the C-C cleavage, rather, it is reasonably explained by a combined experiment and theoretical approach.

Reviewer #2: The authors have satisfactorily addressed my comments to the initial version of manuscript. I have no objection to the publication of this revised manuscript, except for one suggestion: perhaps the authors can consider including the NMR analysis into the manuscript/SI. I suppose this might be useful for some readers.

Our Response: Thank you for your recognition of our work. We have displayed the NMR analysis in the Supplementary Fig. 29. according to your suggestion to provide readers in need.

Reviewer #3: The manuscript is well revised according to the comments. Therefore, now the manuscript is acceptable for publication.

Our Response: Thank you for your recognition of our work.

REVIEWERS' COMMENTS

Reviewer #1 (Remarks to the Author):

The authors have addressed all my questions and concerns in this revision.

Response to the Comments on Nature Communications

Manuscript ID: NCOMMS-23-61325B

Reviewer(s)' Comments to Author:

Reviewer #1: The authors have addressed all my questions and concerns in this revision.

Our Response: Thank you for your recognition of our work.